# The scaffold protein p140Cap limits ERBB2-mediated breast cancer progression interfering with Rac GTPase-controlled circuitries

Silvia Grasso[1], Jennifer Chapelle[1], Vincenzo Salemme[1], Simona Aramu[1], Isabella Russo[1], Nicoletta Vitale[1], Ludovica Verdun di Cantogno[2], Katiuscia Dallaglio[3], Isabella Castellano[2], Augusto Amici[4], Giorgia Centonze[1], Nanaocha Sharma[1], Serena Lunardi[1], Sara Cabodi[1], Federica Cavallo[1], Alessia Lamolinara[5], Lorenzo Stramucci[5], Enrico Moiso[1], Paolo Provero[1], Adriana Albini[6], Anna Sapino[7], Johan Staaf[8], Pier Paolo Di Fiore[9,10,11], Giovanni Bertalot[9], Salvatore Pece[9,11], Daniela Tosoni[9], Stefano Confalonieri[9,10], Manuela Iezzi[5], Paola Di Stefano[1], Emilia Turco[1,*] & Paola Defilippi[1,*]

The docking protein p140Cap negatively regulates tumour cell features. Its relevance on breast cancer patient survival, as well as its ability to counteract relevant cancer signalling pathways, are not fully understood. Here we report that in patients with *ERBB2*-amplified breast cancer, a p140Cap-positive status associates with a significantly lower probability of developing a distant event, and a clear difference in survival. p140Cap dampens ERBB2-positive tumour cell progression, impairing tumour onset and growth in the NeuT mouse model, and counteracting epithelial mesenchymal transition, resulting in decreased metastasis formation. One major mechanism is the ability of p140Cap to interfere with *ERBB2*-dependent activation of Rac GTPase-controlled circuitries. Our findings point to a specific role of p140Cap in curbing the aggressiveness of *ERBB2*-amplified breast cancers and suggest that, due to its ability to impinge on specific molecular pathways, p140Cap may represent a predictive biomarker of response to targeted anti-ERBB2 therapies.

[1] Department of Molecular Biotechnology and Health Sciences, University of Torino, 10126 Torino, Italy. [2] Department of Medical Sciences, University of Torino, 10126 Torino, Italy. [3] Research Infrastructure, IRCCS Arcispedale Santa Maria Nuova, 42100 Reggio Emilia, Italy. [4] Department of Bioscience and Veterinary Medicine, University of Camerino, 62032 Camerino, Italy. [5] Department of Medicine and Aging Science, Center of Excellence on Aging and Translational Medicine (CeSi-Met), G. D'Annunzio University, Chieti-Pescara, 66100 Chieti, Italy. [6] Scientific and Technology Pole, IRCCS MultiMedica, 20100 Milan, Italy. [7] Candiolo Cancer Institute - FPO, IRCCS, Str. Prov. 142, km 3.95, I-10060 Candiolo (To), Italy. [8] Division of Oncology and Pathology, Department of Clinical Sciences, Lund University, Lund 22100, Sweden. [9] Molecular Medicine Program, European Institute of Oncology, 20100 Milan, Italy. [10] IFOM, The FIRC Institute for Molecular Oncology Foundation, 20100 Milan, Italy. [11] Department of Oncology and Hemato-oncology, University of Milan, 20100 Milan, Italy. * These authors contributed equally to this work. Correspondence and requests for materials should be addressed to P.D. (email: paola.defilippi@unito.it).

Breast cancer is one of the most common cancers with greater than 1,300,000 cases and 450,000 deaths each year worldwide[1,2]. Clinically, breast cancer is classified into three basic therapeutic groups: the oestrogen receptor (ER)-positive group, the ERBB2 (also called HER2)-positive group, and the triple-negative breast cancers (TNBCs, also called basal-like), lacking expression of ER, progesterone receptor (PR) and ERBB2 (ref. 2).

The ERBB2 oncogene (the human V-Erb-B2 Avian Erythroblastic Leukemia Viral Oncogene Homolog 2) is a tyrosine kinase receptor, which belongs to the ERBB family. ERBB2 gene amplification and receptor over-expression are causally linked to oncogenesis in ~20% of breast cancers and define a molecular breast cancer subtype characterized by an adverse clinical outcome[3–5]. ERBB2 amplified tumours are a biologically non-homogeneous subgroup of breast cancers[6]. Indeed, although the ERBB2 gene is located in the most highly rearranged segment in chromosome 17 (17q12-q21)[7], the amplification of the surrounding genomic region is a highly variable process that leads to a complex pattern of amplicons. The genes included in the amplicons may significantly contribute to ERBB2 tumour progression and treatment efficacy[7–11].

ERBB2 tyrosine kinase activation at the plasma membrane triggers key signalling pathways that direct general tumorigenicity, including escape from apoptosis, increased cell proliferation and migration, and epithelial to mesenchymal transition (EMT)[12–15].

We have previously described the p140Cap adaptor protein as a molecule that interferes with adhesion properties and growth factor-dependent signalling, thus affecting tumour features in breast cancer cells[16–19]. Recent reports have underlined that p140Cap regulates proliferation and migration in colon, lung, gastric, cutaneous squamous carcinoma and osteosarcoma cancer cells[19–24]. Indeed, in a cohort of breast cancer patients, p140Cap expression was linked to a less aggressive breast cancer disease[25], leading to the hypothesis that in these tumours p140Cap may counteract tumour fitness. However, it was not possible to assess the relevance of p140Cap expression for patient survival in that cohort[25], thus leaving open the question of the relevance of p140Cap to breast cancer prognosis.

In this work, we set out to tackle the relevance of p140Cap in human breast cancer by analysing a large consecutive cohort of patients with invasive breast cancer and we demonstrated a strong association between p140Cap and improved survival of ERBB2 patients. We also found that the p140Cap coding gene, SRCIN1, located on Chromosome 17, one million base pair centromeric from the ERBB2 gene, is amplified together with ERBB2, in >60% of ERBB2 patients. We took advantage of the NeuT mouse model of mammary breast cancer, and of human ERBB2 breast cancer cells, to address the role of p140Cap protein in the ERBB2-related breast cancer disease. Altogether, our results argue for a key role of p140Cap in curbing the aggressiveness of the ERBB2 tumours, counteracting in vivo tumour growth, epithelial mesenchymal transition and metastatic lesions.

## Results

**Decreased metastatic risk in ERBB2 tumours expressing p140Cap.** In a previous study, we showed that p140Cap expression was linked to a less aggressive breast cancer disease[25]. However, the lack of complete clinical follow-up for the cohort used in that study did not allow to assess the prognostic relevance of p140Cap expression in breast cancer. Here, we analysed p140Cap expression, by immunohistochemistry (IHC), on a

consecutive cohort of 622 invasive breast cancers available in a tissue microarray (Supplementary Table 1). Data for p140Cap expression were available for 515 out of 622 samples (Fig. 1a; Supplementary Table 2). Positive p140Cap status (IHC score ≥1) was associated with good prognosis markers, such as negative lymph node status ($P = 0.014$, where $P$ = Pearson $\chi^2$-test), ER and progesterone receptor (PgR)-positive status ($P = 0.0002$ and $P = 0.0049$, respectively), small tumour size (pT1 versus pT2–pT4, $P < 0.0001$), low grade ($P < 0.0001$), low proliferative status (Ki67, $P = 0.0013$), and ERBB2-positive status ($P = 0.0344$). Positive p140Cap status was also associated to breast cancer molecular subtypes, being expressed in >85% of Luminal A tumours, 77% of Luminal B, and only 56% of triple-negative tumours (Supplementary Table 2).

In univariate analysis, a positive p140Cap status was associated with a lower risk of developing distant metastases, and of death from breast cancer in the entire breast cancer cohort (Fig. 1b). However, a more in-depth analysis revealed that the prognostic effect of p140Cap in the consecutive cohort of breast cancer patients was to be ascribed to its performance in the subgroup of ERBB2-amplified breast cancers (Fig. 1c), in which a high p140Cap status predicts a significantly lower probability of developing a distant event (left panel), and a clear difference in survival (right panel). By contrast, no significative differences could be observed in patients not harbouring ERBB2 amplification (Fig. 1d). The prognostic power of p140Cap was lost in a multivariate analysis, indicating that p140Cap is not an independent prognostic marker in breast cancer (Supplementary Fig. 2A; Supplementary Table 2). However, in the ERBB2-amplified subgroup, the lymph node status was the sole independent predictive marker, in multivariate analysis. When this group of tumours was subjected to a bivariate analysis, with nodal status and p140Cap expression, the two variables were found to be independent of each other in their association with prognostic outcome (Supplementary Fig. 2B).

In conclusion, p140Cap expression associates with reduced risk of metastasis (and death from cancer), in the ERBB2-amplified subgroup of breast cancer patients, arguing for a possible role of p140Cap in counteracting the migratory and/or metastatic ability of ERBB2-amplified tumour cells.

**SRCIN1 is co-amplified with ERBB2 in ERBB2 amplified patients.** p140Cap is encoded by the SRCIN1 gene, located at Chromosome 17q12, one million base pair centromeric to the ERBB2 gene. Several genes included in the amplicons have been reported to play a role in ERBB2 tumour progression[7–11]. However, the co-amplification of SRCIN1 gene in the context of the ERBB2-related disease has not yet been deeply investigated.

To assess how frequently SRCIN1 gene may be included in the ERBB2 amplicon, BAC array Comparative Genomic Hybridization (aCGH) was performed. The analysis of 200 ERBB2-amplified tumours from a large Swedish Cohort[8], showed that the SRCIN1 gene is altered in 70% of cases, with 123 cases (61.5% of the total) showing a copy number (CN) gain for SRCIN1 (Fig. 2a). Kaplan–Meier analysis of these tumours showed that SRCIN1 amplification correlates with significantly improved survival (Supplementary Fig. 3). Moreover, mRNA expression and SRCIN1 gene CN from 50 of the 200 ERBB2 amplified tumours were significantly correlated, giving a Pearson correlation of 0.77 (Fig. 2b).

Further, we analysed by FISH a consecutive series of 77 breast cancer patients at diagnosis with a mix of probes for SRCIN1 and the centromeric region (CEP17) of chromosome 17. While in 43 ERBB2-negative breast cancers SRCIN1 CN was never altered, in ERBB2-amplified tumours[26], 56% of the specimens were

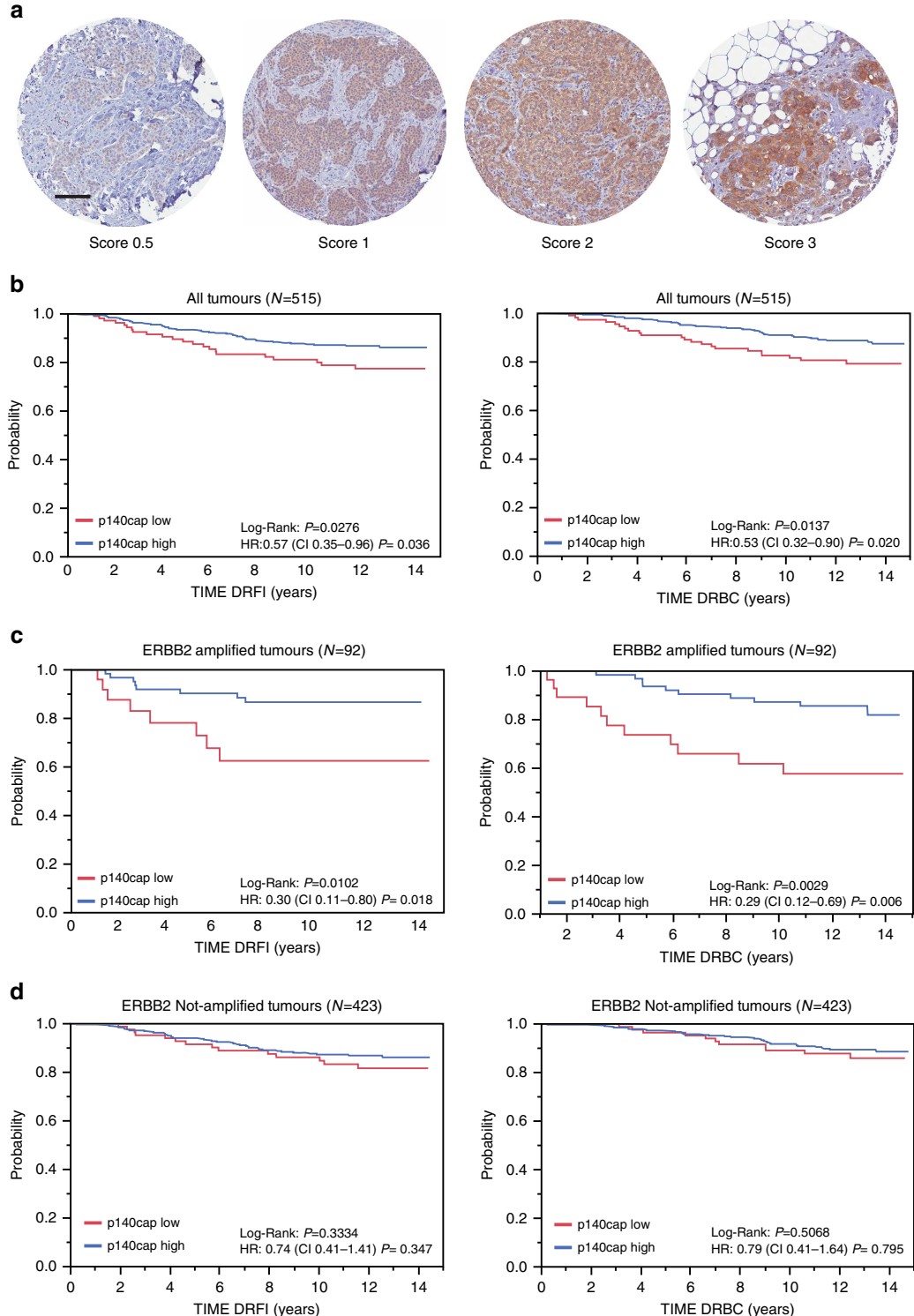

**Figure 1 | Prognostic relevance of p140Cap expression in breast tumours.** (**a**) p140Cap expression was measured by IHC on tissue microarray (TMA) samples. For the purpose of correlation with clinical and pathological parameters, tumours were classified based on the intensity of p140Cap staining as 0.5–3: p140Cap-Low (IHC score <1) and p140Cap-High (IHC score ≥1). Images are representative of p140Cap expression scoring according to intensity staining in TMA. In tumour tissues, the IHC signals were associated with the tumour cell component and not with the adjacent or infiltrating stroma. TMA data analysis was performed using JMP 10.0 statistical software (SAS Institute, Inc). Scale bar, 100 μm. (**b**) p140Cap expression in the whole cohort: Distant Recurrence Free Interval (DRFI)[65] (left panel: hazard ratio: 0.57, $P = 0.036$); and Death Related to Breast Cancer (DRBC; right panel: hazard ratio: 0.53, $P = 0.020$). (**c**) p140Cap expression in ERBB2-positive patients: DRFI (left panel: hazard ratio: 0.30, $P = 0.018$); and DRBC (right panel: hazard ratio: 0.29, $P = 0.006$). (**d**) p140Cap expression in ERBB2-negative patients: DRFI (left panel: hazard ratio: 0.74, $P = 0.347$); and DRBC (right panel: hazard ratio: 0.41, $P = 0.795$). $P$ = Pearson $\chi^2$-test.

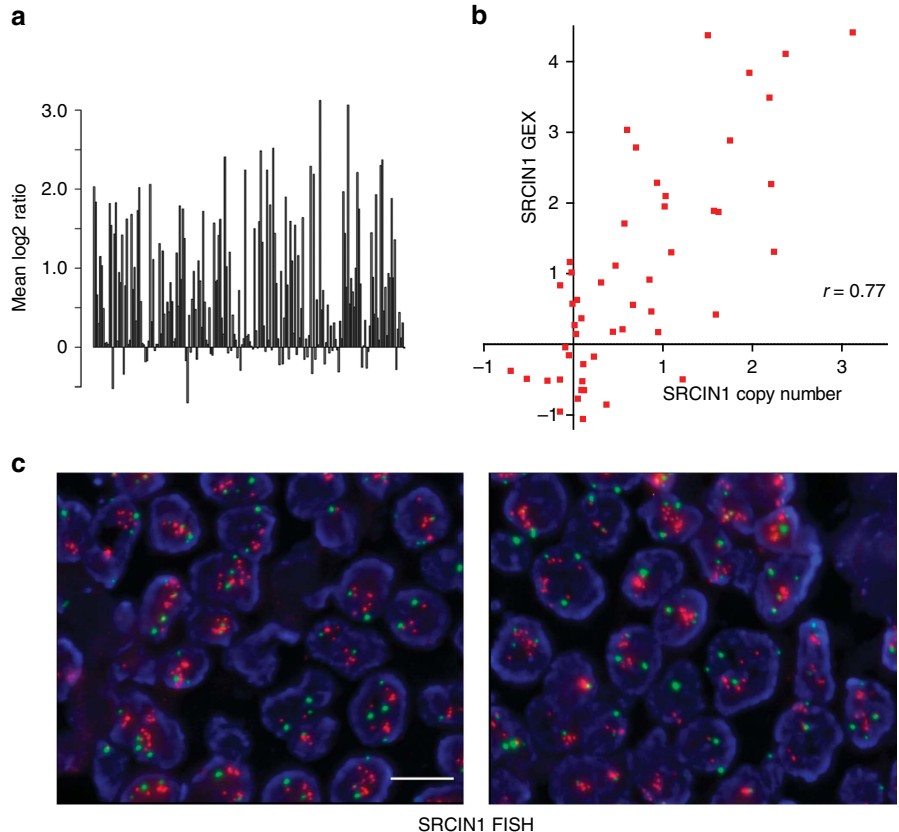

**Figure 2 | *SRCIN1* gene alterations in human ERBB2 breast cancer samples.** (**a**) *SRCIN1* gene copy number across 200 *ERBB2*-amplified breast cancer samples analysed by aCGH. *Y* axis corresponds to log2 transformed copy number, where values $> 0$ correspond to increased copy numbers, and values $< 0$ to copy-number loss. Bars represent individual samples. (**b**) Correlation of *SRCIN1* gene expression (GEX; *y* axis) and *SRCIN1* gene copy number (*x* axis) for 50 ERBB2 amplified cases from ref. 6. To assess whether this increase in *SRCIN1* gene copy number results in increased mRNA expression, gene expression data were compared with aCGH log2ratios using the Pearson correlation as described in ref. 61. Pearson's coefficient of correlation is 0.77. (**c**) p140Cap FISH of breast cancer tissues. Representative images of two cases of *ERBB2* amplified tissues, labelled with a mix of two probes *SRCIN1*/CEP1; Red (*SRCIN1*) and green (CEP17) spots were automatically acquired at 40X, using Metafer, by a MetaSystem scanning station. left panel: 95% *SRCIN1* amplification; average *SRCIN1*/nuclei = 11.7; right panel: 90% *SRCIN1* amplification; average *SRCIN1*/nuclei = 13,4. Scale bar, 10 μm.

amplified for SRCIN1 (Fig. 2c). These data indicate that alterations at the level of the *SRCIN1* locus are strictly linked to chromosomal rearrangements that result in *ERBB2* amplification. Altogether, these results show that the *SRCIN1* gene is frequently, but not obligatorily, co-amplified with *ERBB2* in breast cancers, arguing for a potential role of *SRCIN1* as a determinant of the clinical heterogeneity of ERBB2 tumours. These observations also provided us with the testable hypothesis that the presence of *SRCIN1* may attenuate the intrinsic biological aggressiveness of breast tumours with *ERBB2* alterations.

**p140Cap limits tumorigenicity of NeuT-driven breast tumours.** To test the above hypothesis, we generated a transgenic (Tg) mouse model in which p140Cap expression is driven under the control of the MMTV promoter (MMTV-p140Cap; Fig. 3a), to cross them with a well-characterized model of ERBB2-dependent breast carcinogenesis, the Tg MMTV-NeuT mouse model[27,28]. We selected two MMTV-p140Cap lines with a strong p140Cap expression in the mammary gland (see Supplementary Fig. 4 for detailed characterization of the Tg mice) that were crossed with both FVB-MMTV-NeuT[29] and BALB/c-MMTV-NeuT[27,28] mice, which display different tumour onset times, to generate p140-NeuT mice. p140Cap expression in tumours derived from these mice was confirmed by Western blot analysis (Fig. 3b). When compared to either FVB-NeuT or BALB/c-NeuT mice, the

corresponding p140-NeuT mice showed a significant delay in the appearance of the first tumour (Fig. 3c, Fisher's exact test, Two sided, $P = 0.0022$; $P = 0.0056$) associated with a significant decrease in the total tumour burden (Fig. 3d, unpaired *t*-test: $P < 0.001$, $P < 0.05$). Histological analyses showed morphological differences in the appearance of the two types of tumours (Fig. 3e). NeuT tumours were composed of large solid nodules, separated by delicate bundles of stromal tissue, with necrosis often evident in the centre of the largest nodules (Fig. 3e, panels a,b). Tumours developed in p140-NeuT mice consisted of smaller nodules and sheets of cells separated by more abundant stroma, with cancer cells extending into the stroma in nest-like formations showing distinctive holes between the cancer cells (Fig. 3e, panels c,d). Both tumour types were strongly positive for NeuT ((Fig. 3e, panels e–h) and for cytokeratins CK8/18 (see Supplementary Fig. 5A). A larger percentage of NeuT tumour cells were positive for the proliferation marker PCNA (Fig. 3e, panels i,j), compared to p140-NeuT tumour cells (Fig. 3e, panels k,l). PCNA quantification is shown on the right of Fig. 3e ($32 \pm 1,560$ versus $18,65 \pm 2,141$). Angiogenic infiltration, as assessed by CD31 marker staining, was also decreased in p140 tumours ($9,648 \pm 351.5$ versus $5,344 \pm 232.8$; Supplementary Fig. 5B). Not significant differences were detectable in activated Caspase3 staining, in which only a few cells were positive in both tumour types ($7,694 \pm 2,257$ versus $7,381 \pm 2,408$;

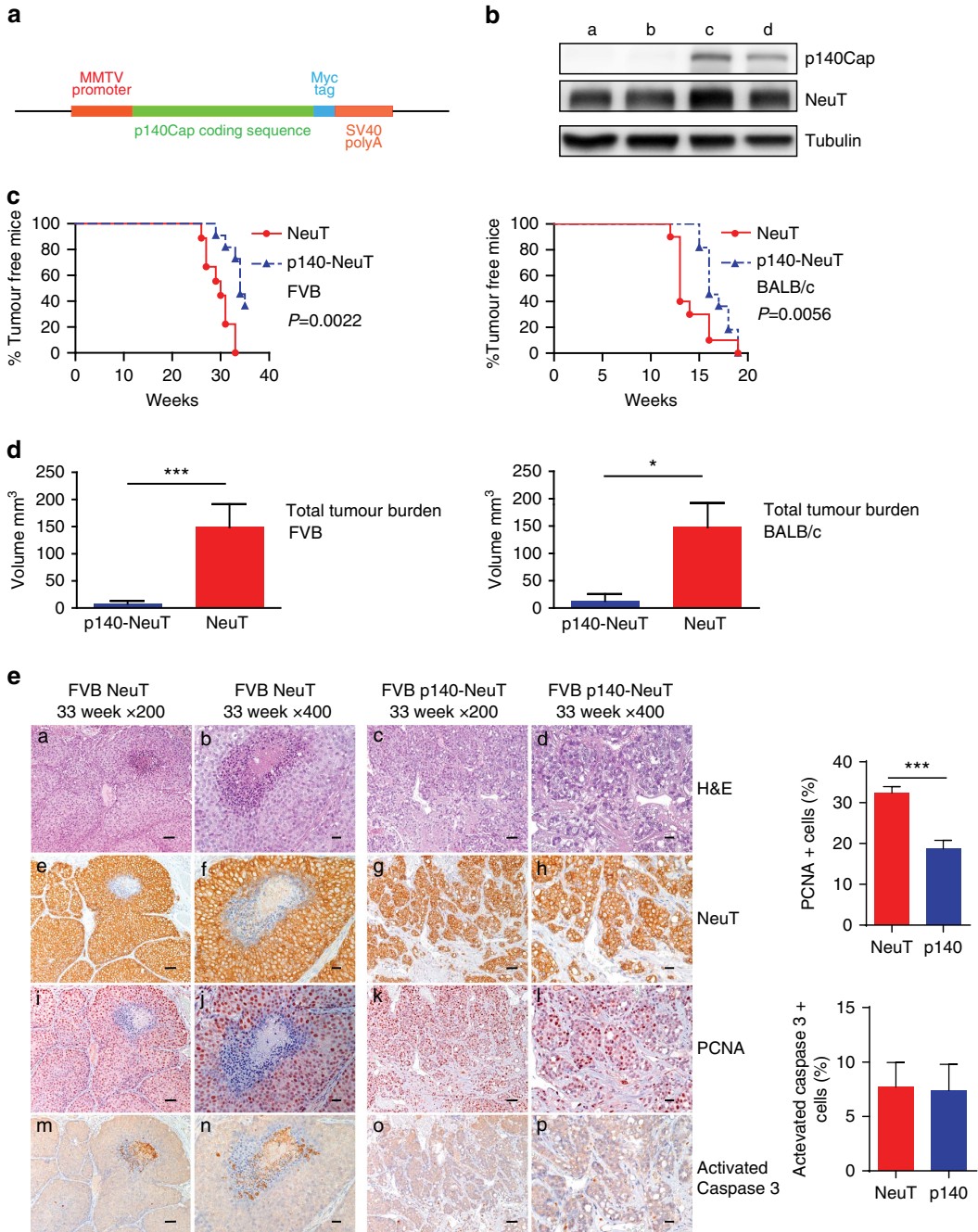

**Figure 3 | p140Cap expression is causative in limiting tumour growth in NeuT mice.** (**a**) Expression cassette used for the generation of MMTV-p140Cap transgenic mice. The Myc epitope is inserted at the carboxyterminal region of the protein. (**b**) Extracts of tumours derived from NeuT mice (a,b) or p140-NeuT mice (c,d) were run on 6% SDS–PAGE and stained with antibodies to p140Cap, ERBB2 and tubulin for loading control. (**c**) Percentage of tumour free mice in NeuT (red line) and p140-NeuT (blue-dashed line) transgenic animals in both FVB (left) or BALB/c background (right). Twelve mice were analysed for each group. Fisher's exact test, Two sided, $P = 0,0022$; $P = 0,0056$. Error bar: s.e.m. (**d**) Total tumour burden in NeuT (red) and p140-NeuT (blue) mice in both the FVB (left) or BALB/c backgrounds (right) was measured. Ten mice were analysed in each group. Unpaired $t$ test: (*$P < 0.05$; ***$P < 0.001$). Error bar: s.e.m. (**e**) Paraffin-embedded sections from three NeuT and three p140-NeuT tumours taken from mice at 33 weeks of age were analysed for hematoxylin–eosin (H&E) (a–d) and for immunohistochemistry with antibodies to NeuT (e–h), PCNA (i–l) and activated Caspase3 (m–p). Representative images are shown. Scale bar, 50 μm (first and third columns); 20μm (second and fourth columns). Histograms on the right show the percentage of PCNA+ (upper panel) and Activated Caspase3+ (lower panel) cells. Statistical significative differences were evaluated using unpaired $t$-tests (***$P < 0.001$).

Fig. 3e, panels m–p). Activated Caspase3 quantification is shown on the right of Fig. 3e. Altogether, these results show that p140Cap expression attenuates the phenotype of NeuT tumours *in vivo*, resulting in the development of smaller and lower grade mammary carcinomas.

**p140Cap reverts the NeuT effects on mammary morphogenesis.** It is well known that activation of the ERBB2 oncogene is sufficient to disrupt the morphogenetic program that drives the formation of the mammary gland acini *in vitro*[30,31]. Indeed, normal mammary epithelial cells embedded into three-

dimensional (3D) Matrigel-Collagen cultures give rise to hollow glandular-like acini displaying features of luminal differentiation[32–34]. In contrast, ERBB2 transformed cells escape apoptosis responsible for the cavitation process and originate aberrant filled-type structures, a phenotype linked to the cellular transformation with loss of apical–basal polarity[30,31]. On the basis of our evidence that p140Cap is able to curb NeuT-driven tumorigenesis *in vivo*, we set out to evaluate whether p140Cap also counteracts the disruption of the mammary morphogenetic program caused by ERBB2. To this aim, we established primary

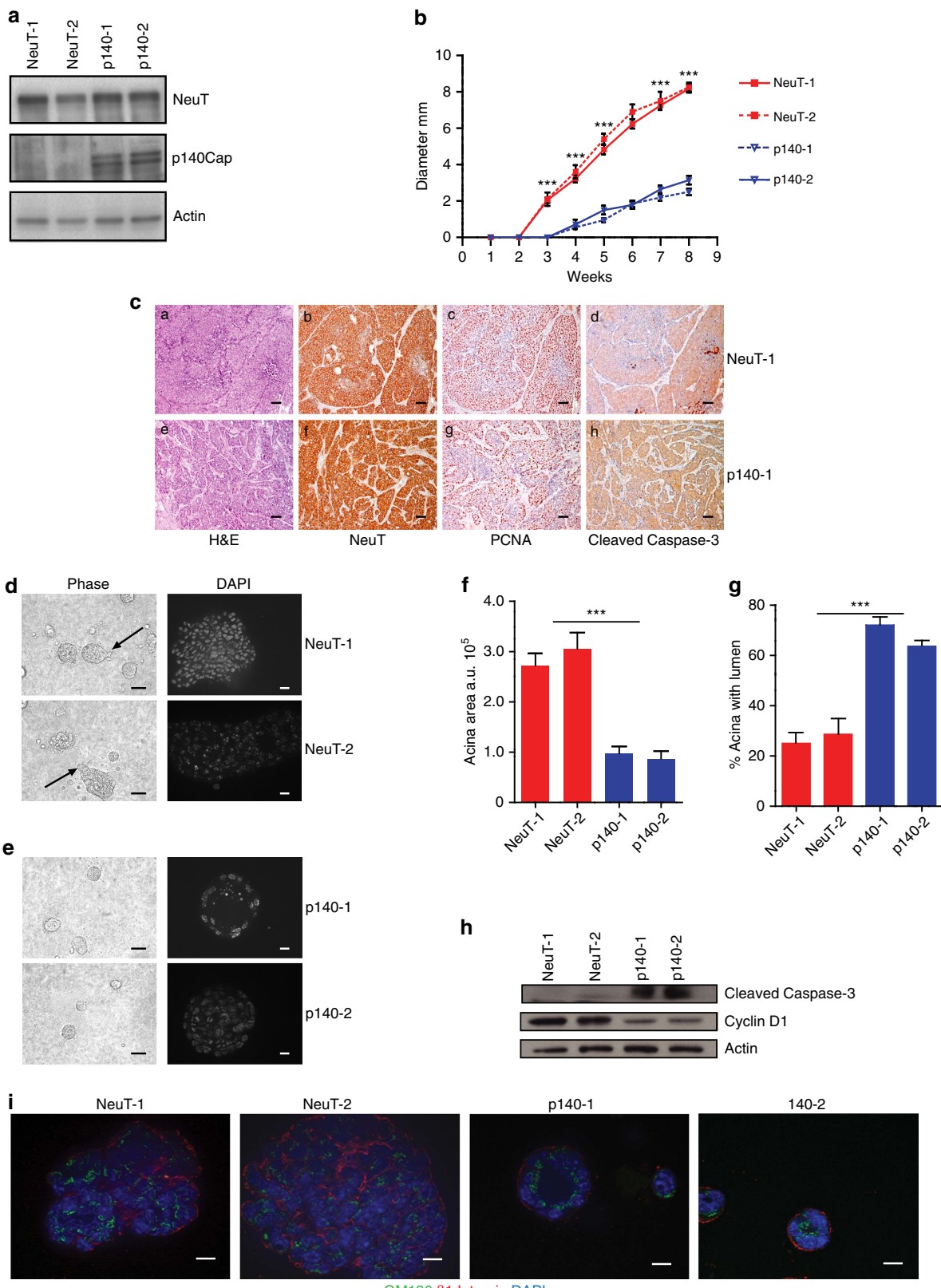

epithelial cancer cells from NeuT and p140-NeuT tumours in the BALB/c background. Two populations for each genotype, which expressed comparable levels of NeuT (NeuT-1 and NeuT-2) or p140Cap (p140-1 and p140-2; Fig. 4a), were chosen for further experiments. The cell lines, even if in standard culture condition did not show difference in proliferation (Supplementary Fig. 6A), displayed a distinct behaviour in apoptosis assays. Indeed, p140Cap primary tumour cells showed increased percentage of cells expressing the apoptosis marker Annexin 5 and increased expression of cleaved Caspase 3, when subjected to apoptotic stimuli, such as starvation or matrix cell detachment in culture conditions (Supplementary Fig. 6B). Moreover, they retained in transplantation assays the characteristics of the parental tumours, as evidenced by reduced tumour burden (Fig. 4b) and a more differentiated appearance of tumours (Fig. 4c), comparing p140 to NeuT cells.

When these cells were plated in 3D Matrigel-Collagen cultures for 15 days, NeuT-1 and NeuT-2 cells yield large multi acinar, apolar structures of irregular shape (Fig. 4d) that sometimes displayed protrusions (see arrows), suggestive of invasive features. These structures frequently did not show a lumen. In contrast, p140-1 and p140-2 cells formed smaller acinar-like structures with regular borders, without protrusions, which frequently displayed a lumen (Fig. 4e). Figure 4f,g show a quantification of these two phenotypes. Biochemically, these events were measured via the decreased expression of the proliferation marker Cyclin D1 (Fig. 4h), which is consistent with the decreased PCNA levels detected in p140-NeuT tumours *in vivo*. Moreover, p140Cap expression resulted in a substantial increase in the levels of activated Caspase3 in 3D cultures (Fig. 4h). Overall, in 3D conditions, p140Cap restores the dynamic equilibrium between cell proliferation and cell death which is typical of normal mammary epithelial cells during tissue morphogenesis[35].

The observed morphological features were mirrored by restoration of apical–basal polarity. Polarity properties were dissected in cells grown in 3D Matrigel-Collagen cultures for 15 days, via staining with the apical Golgi marker GM130 and the basal marker beta1 integrin[30]. The structures formed by the NeuT cells showed loss of Golgi marker GM130 orientation towards the lumen, and beta1 integrin mis-localization (Fig. 4h). In contrast, in p140 cells, GM130 always localized in the inner part of the acini, oriented towards the lumen, while beta1 integrin was clearly restricted in the outer part of the acini, to define the basal compartment (Fig. 4i). Thus, at least under the conditions of *in vitro* assays, the mitigating effect of p140Cap on ERBB2 tumour growth could be correlated with the re-enactment by p140Cap of the differentiation program disrupted by ERBB2.

**p140Cap limits EMT in the NeuT cells.** EMT is integral to several steps of the metastatic process[15,36]. In keeping with this, we found that the presence of p140Cap was associated with a marked down-regulation of an EMT transcription program, as witnessed by the significantly reduced expression of mRNA transcripts for the EMT transcription factors Snail, Slug and Zeb1 (Fig. 5a), and for the mesenchymal cell-cell adhesion protein N-cadherin in p140 cells compared to NeuT cells. Consistent with these findings, p140 tumour cells also displayed up-regulation of the mRNA levels for the epithelial E-cadherin mRNA (Fig. 5b). The overall inhibitory effect of p140Cap on EMT was further confirmed by western blot analysis that showed, in p140 versus NeuT cells, reduced expression levels of Snail and N-cadherin proteins combined with increased levels of E-cadherin (Fig. 5c). Immunofluorescence staining of E-cadherin on tumour sections (Fig. 5d), confirmed the increased expression of E-cadherin at the cell membrane in p140 tumours compared to NeuT tumours. Altogether, the sum of these data argues that p140Cap may effectively decrease pathways related to the progression of ERBB2 tumours, contributing to increased patient survival.

**p140Cap limits metastasis in NeuT expressing cells.** On the basis of the association between p140Cap status and reduced risk of distant metastasis in ERBB2 breast cancer patients, and on the down-regulation of the EMT transcription program, we addressed the putative protective role of p140Cap against the metastatic risk. In a spontaneous metastasis assay from primary tumours, we did not detect lung metastasis from neither NeuT or p140 xenotransplants. To address this point, we moved to the NeuT-TUBO cells, an additional transplantable primary NeuT cell model derived from a tumour arisen in BALB/c-MMTV-NeuT mice[37]. Upon infection with empty or p140Cap retroviruses, we generated NeuT-TUBO (as mock cells), and p140-TUBO cells (Supplementary Fig. 7A). We showed that p140Cap expression significantly limited tumour cell growth upon transplantation (Supplementary Fig. 7B). In the experimental metastasis assays upon tail vein injection, NeuT-TUBO cells gave rise, after 25 days, to numerous large lung metastases substituting ~80% of lung tissue area. At the same time point, p140-TUBO cells were grown to occupy only ~54% of lung tissue area (Fig. 6a). Since this assay is only a proxy to measure the metastatic potential of cells, we moved to the spontaneous metastasis assay from primary tumours, comparing metastasis formation from tumours of the same size. As shown in Fig. 6b, tumours originated from p140-TUBO cells gave rise to a significantly reduced number of lung metastases over tumours grown from NeuT-TUBO cells.

**Figure 4 | Primary p140 cancer cells restore mammary epithelial acina morphogenesis in 3D Matrigel-Collagen cultures.** (**a**) Protein extracts from two independent primary cancer cells for each genotype (NeuT-1, NeuT-2, p140-1 and p140-2) were run on 6% SDS–PAGE and stained with antibodies to NeuT, p140Cap and actin for loading control. (**b**) $10^6$ cells as in **a** were injected in the left and right fat pads of nude mice. Tumour diameters were measured every week for 8 weeks. Two independent experiments were performed using five mice per group. Differences in tumour diameter were evaluated using two-way analysis of variance (ANOVA) followed by Bonferroni multiple comparison *post hoc* tests (***$P < 0.001$). (**c**) Paraffin-embedded sections were prepared at the end of the experiments from tumours derived from mice as in **b**. Sections were analysed for Hematoxylin–Eosin (a,e), and for immunohistochemistry with antibodies to NeuT (b,f), PCNA (c,g) and activated Caspase3 (d,h). Representative images are shown. Scale bar, 50 μm. (**d,e**) Primary cancer cells for each genotype (NeuT-1, NeuT-2, p140-1, and p140-2) were plated in Matrigel/Collagen I 1:1 and left to grow for 15 days. Day 15 acina are shown as phase images in the left panels, or as Dapi nuclei staining (bright grey) in right panels. Arrows indicate the presence of invasive protrusions. Representative images from three independent experiments are shown. Scale bar, 50 μm. (**f**) The histogram represents the area of the acina quantified by the computer-generated software Zeiss Axiovision 4.5 and shown in arbitrary units (a.u.). (**g**) The histogram represents the percentage of acina structures with an internal lumen. The lumen has been manually quantified. In **f** and **g**, statistical significative differences were evaluated using unpaired *t* tests. Error bar: s.e.m. (***$P < 0.001$). (**h**) Primary cancer cells as in d were plated in Matrigel/Collagen I 1:1 and left to grow for 12 days. Protein extracts were run on 4–12% SDS–PAGE and stained with antibodies to Cleaved Caspase 3, Cyclin D1 and Actin for loading control. (**i**) Primary cancer cells as in **d** were analysed as day 15 acinar structures by immunostaining for a *cis*-Golgi matrix protein, GM130 (green), and a basal marker protein, beta1 integrin (red). Nuclei were co stained with DAPI (blue). Representative images are shown. Scale bar, 50 μm.

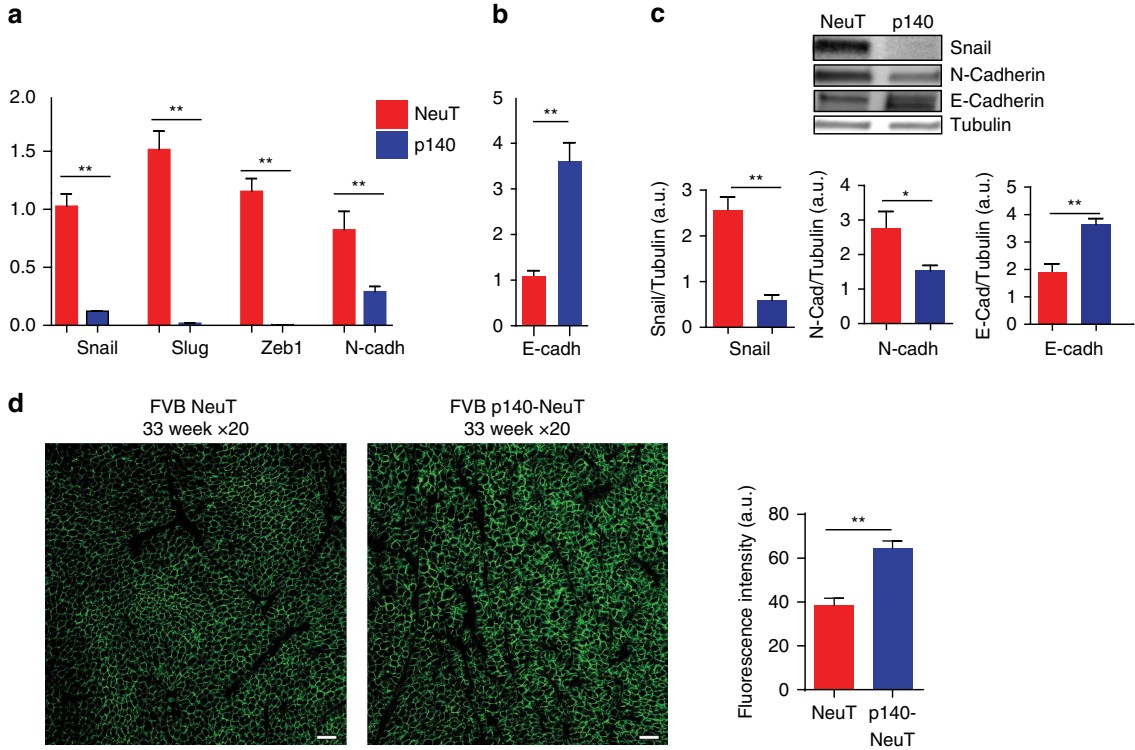

**Figure 5 | Primary p140 cancer cells show impaired progression features. (a,b)** RT–PCR was performed to quantify mRNA expression of EMT markers in NeuT and p140 cancer cell RNA. RNA was prepared from three biological replicates. RT–PCR was performed on triplicates. CNRQ (calibrated normalized relative quantity) is shown in arbitrary units (a.u.). Error bar: s.e.m. **(c)** Protein extracts from NeuT and p140 cells were run on 4–15% SDS–PAGE and stained with antibodies to Snail, N-cadherin and E-cadherin. Tubulin was used as an internal standard for protein loading. Histograms show in a.u. the quantification of three independent experiments. Statistical significative differences were evaluated using unpaired t tests (*$P < 0.05$; **$P < 0.01$; ***$P < 0.001$). Error bar: s.e.m. **(d)** Paraffin-embedded sections from three NeuT and three p140-NeuT tumours taken from mice at 33 weeks of age were analysed for immunofluorescence with antibodies to E-Cadherin (green). Representative images are shown. Scale bar, 50 μm. Image acquisition was performed using Zeiss LSM 510 META confocal microscope. The E-cadherin mean fluorescence intensity was evaluated on the digital images of three tumours per group (4 × 200 microscopic fields per sample) with ImageJ, using the Mean Gray Value.

Moreover, the immunohistochemical staining with anti NeuT/ERBB2 and anti p140Cap antibodies of lungs with p140-TUBO metastases showed that while all the metastases were positive for the NeuT protein, p140Cap strong expression was conserved only in small metastases (Fig. 6c). Interestingly, larger metastases expressed only low levels of p140Cap and, accordingly, showed a less nodular histological structure similar to those developed from NeuT-TUBO tumours (Fig. 6d, compare upper and lower panels). Overall, these data indicate that p140Cap counteracts metastasis formation.

**p140Cap attenuates ERBB2-driven Rac-dependent circuitries.** The sum of results from (i) the analysis of the clinical cohort (reduction of metastatic risk in ERBB2 tumours), (ii) the experiments performed *in vivo* (reduced tumour masses, reduced 'metastatic' ability and decreased expression of EMT markers in the animal model) and (iii) *in vitro* (reduced protrusive ability of acini and restoration of polarity in the 3D-morphogenetic assays), points to a counteraction of p140Cap on ERBB2-dependent tumour progression. All these data show that p140Cap dampens tumour features, affecting tumour growth, sensitivity to apoptosis and metastatic properties of ERBB2-positive cancer cells. In search of a molecular mechanism, we decided to exploit SKBR3 breast cancer cells as a model of *ERBB2* gene amplification relevant to human breast cancer. In these cells, we both over-expressed and silenced p140Cap, without altering ERBB2

expression (Supplementary Fig. 8A,B), and tested the effects of these perturbations first on migratory abilities. In a transwell assay, migration was significantly decreased in p140Cap-over-expressing cells (oe p140; Fig. 7a) and increased in p140Cap-silenced cells (si p140; Fig. 7c). Increased migration of p140Cap-silenced cells was also observed in MDA-MB-453 breast cancer cells, another model of ERBB2 gene amplification relevant to human breast cancer (Fig. 7e; Supplementary Fig. 8C). Consistently, the migration of murine p140 cells lines, derived from transgenic mice, was profoundly inhibited, when compared to NeuT lines (Fig. 7g).

Furthermore, we examined whether p140Cap can control potential downstream signalling mechanisms. We have already shown that p140Cap can control Src activation[16,19]. Interestingly, both in NeuT and in SKBR3 cells, p140Cap expression did not affect the activation of the Src kinase and the phosphorylation of its effectors, p130Cas and paxillin, compared to MDA-MB-231 (ref. 19; Supplementary Fig. 9), suggesting that in ERBB2 transformed cells, p140Cap acts on additional pathways for the control of cell migration.

Several Rho GTPases are frequently altered in tumours and metastases and this often correlates with poor prognosis[38,39]. In particular, Rac is an essential effector pathway for ERBB2-mediated breast cancer progression to metastasis[40–43]. In SKBR3 cells, inhibition of ERBB2 activation by Lapatinib treatment significantly impaired both ERBB2 phosphorylation on Tyr 1,248 and Rac activation (Supplementary Fig. 10), confirming that

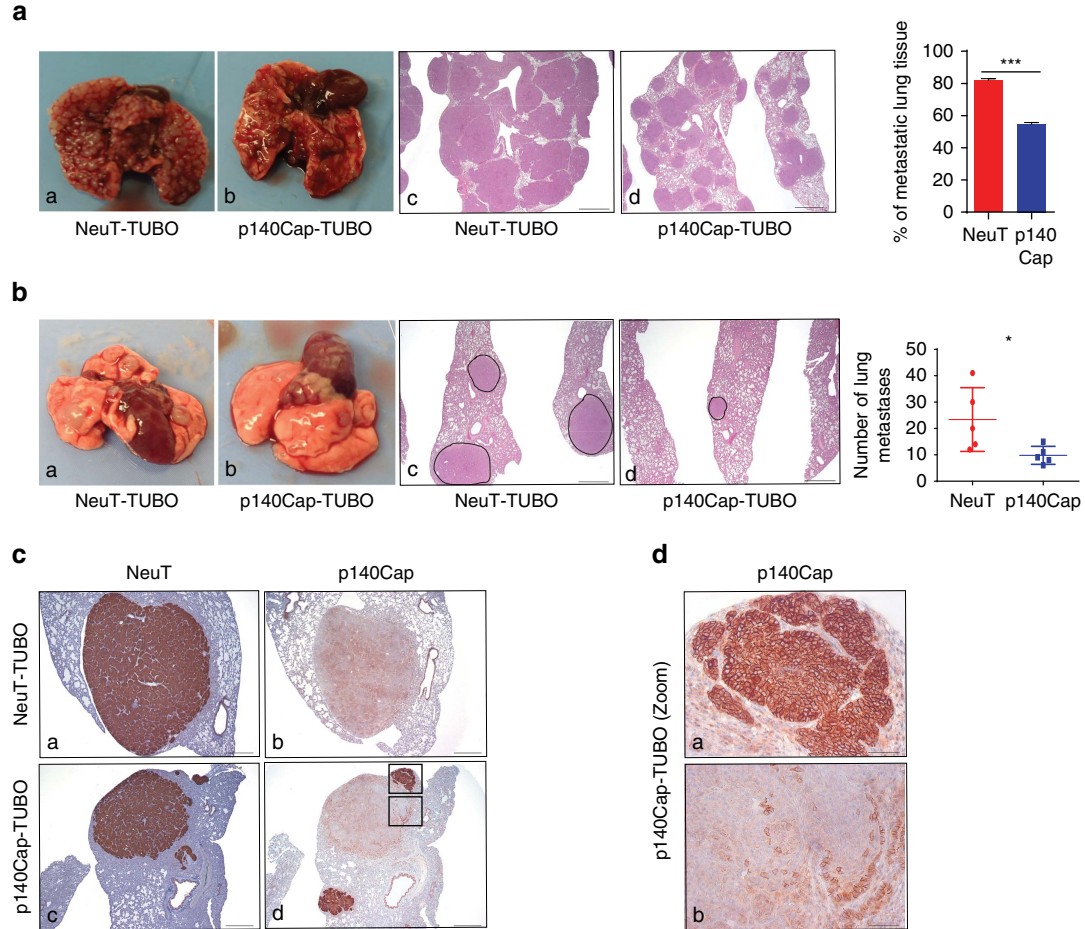

**Figure 6 | p140Cap impairs spontaneous metastasis.** (**a,b**) Representative gross observation (a,b), hematoxylin and eosin sections (c,d) and quantitative analysis in the experimental lung metastasis assay (**a**) and in the spontaneous lung metastasis assay (**b**). In **a**, $5 \times 10^4$ cells were injected into the tail vein of NSG mice. After 4 weeks, the lungs were explanted and analysed. Two independent experiments were performed using five mice per group. The histogram shows the percentage of metastatic lung tissue on total lung area. Statistical significance was evaluated with unpaired $t$-test (***$P < 0.001$). Error bar: s.e.m. In **b**, $10^5$ cells were injected in the right fat pads of NSG mice. Tumour volumes were measured every week; tumours were surgically removed when they reached a 10 mm diameter. After 5 weeks, mice were killed and lungs were explanted. The histogram shows the number of lung metastasis. Statistical significance was evaluated with unpaired $t$-test (*$P < 0.1$). Error bar: s.e.m. Scale bar (a,b): 800 μm. (**c**) NeuT (a,c) and p140Cap expression (b,d) in spontaneous lung metastasis of mice injected with NeuT-TUBO cells (a,b) and mice injected with p140-TUBO cells (c,d). Scale bar, 400 μm. (**d**) High-magnification fields of rectangular areas in **c**, (panel **d**). Scale bar, 50 μm.

Rac is a downstream effector of ERBB2 also in our experimental system[44]. In the same cells, consistent with defective migration, Rac activity was significantly decreased upon p140Cap over-expression (Fig. 7b) or enhanced upon p140Cap silencing (Fig. 7d,f). These data mirrored those obtained in NeuT tumour-derived cells (Fig. 7h; Supplementary Fig. 11A), indicating that p140Cap affects Rac activity in both human and mouse ERBB2 transformed cells.

Treatment of NeuT cells with the Rac inhibitor NSC23766 (ref. 45) phenocopied the effects of p140Cap expression on the 3D morphogenetic program of NeuT cells, yielding acinar structures that were significantly smaller in size compared to those observed in the NeuT cells, and that frequently displayed a polarized phenotype (Fig. 7i). Finally, expression of a constitutively active mutant of Rac (RacV12) into p140 cells (Supplementary Fig. 12), caused a significant increase in the size of acini, accompanied by an almost complete loss in polarity and an enhancement in invasive protrusions (Fig. 7j). This latter set of data shows that Rac is epistatic to p140Cap, a scenario compatible with the possibility that p140Cap is an upstream regulator of Rac.

**p140Cap limits Rac GEF Tiam1 activation in cancer cells**. To probe into the hypothesis that p140Cap may act upstream of Rac, we focused on the Rac specific activator, the Guanine Exchange Factor (GEF) Tiam1, also in light of the fact that the Rac inhibitor NSC23766, which phenocopies p140Cap expression in NeuT cells (Fig. 7i), is a selective inhibitor of the interaction between Tiam1 and Rac[45]. In both p140 and NeuT expressing cells (Fig. 8a; Supplementary Fig. 11B) and in human p140Cap overexpressing SKBR3 cells (Fig. 8b), we found that p140Cap and Tiam1 co-immunoprecipitated, arguing for their physical interaction *in vivo*. We then investigated whether p140Cap could affect Tiam1 activity in both cell systems. This was established by *in vitro* pull-down experiments using as bait GST-RacG15A, a nucleotide-free Rac mutant that selectively interacts with active Tiam1 (ref. 46). Indeed, in p140 cells, we observed a marked decrease in the recovery of activated Tiam1 by GST-RacG15A, in comparison to NeuT expressing cells (Fig. 8c; Supplementary Fig. 11C). These data were mirrored by those obtained in SKBR3 cells, in which Tiam1 activity was significantly decreased upon p140Cap over-expression (Fig. 8d) or enhanced upon p140Cap silencing (Fig. 8e). Increased Tiam1 activity was also observed in

p140Cap-silenced MDA-MB-453 cells (Fig. 8f). Overall, these data indicate that Tiam1 activity is dependent on p140Cap in these cells.

Taken together, these data show that p140Cap interferes with the Rac circuitries that control *ERBB2* tumour progression, by binding to Tiam1, leading to both Tiam1 and Rac inactivation.

## Discussion

We herein show for the first time that the expression of the p140Cap adaptor protein is clinically relevant to the naturally occurring ERBB2-related breast cancer disease. Indeed, ERBB2 patients who display a positive p140Cap status have significantly higher survival rate, with lower probability of

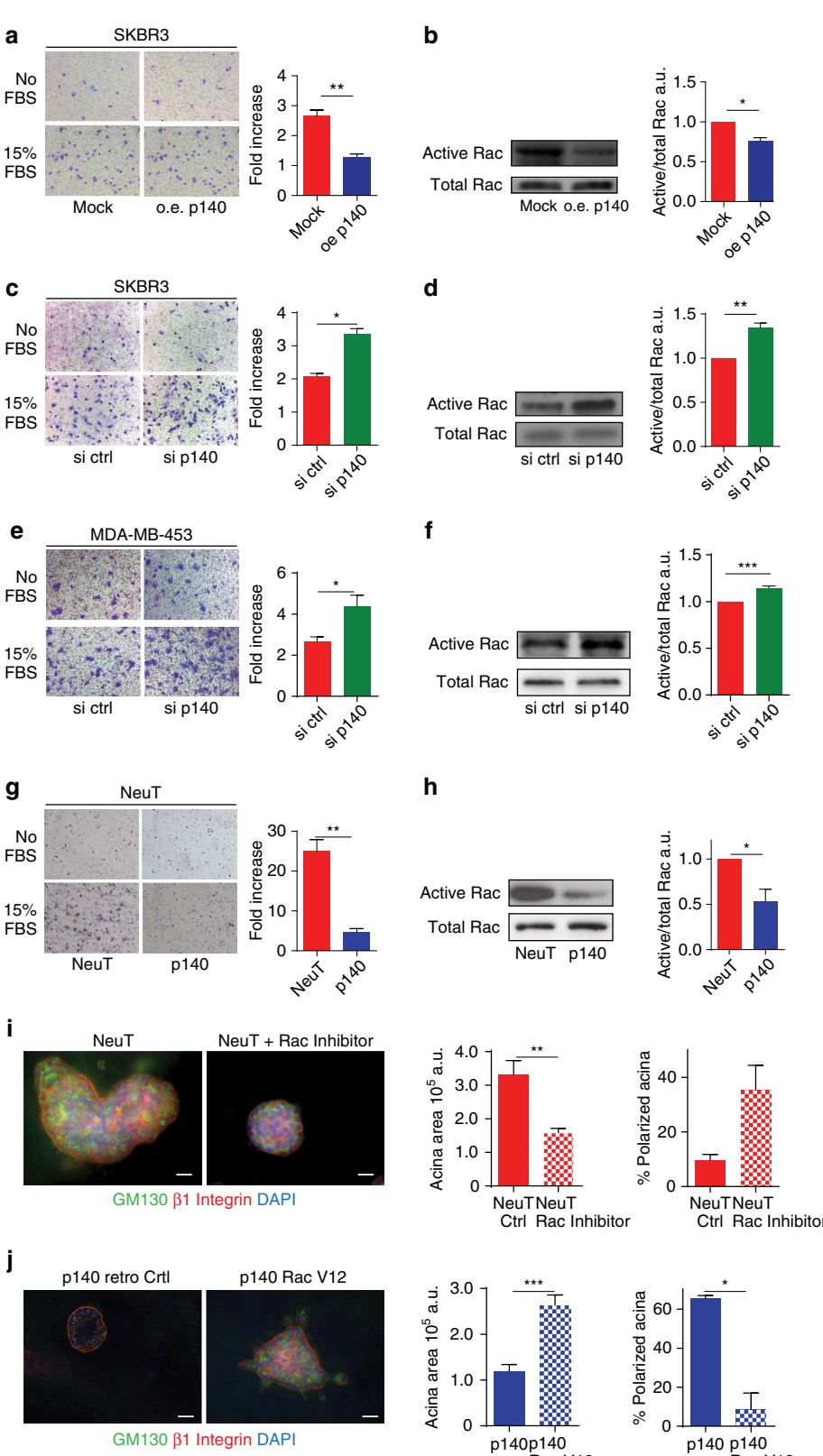

developing a distant recurrence. The clinical evidence that p140Cap correlates with a favourable outcome in ERBB2 breast cancer patients suggest that p140Cap is able to curb the intrinsic biological aggressiveness of ERBB2 tumour (Fig. 8g). Indeed, p140Cap confers to ERBB2 transformed cells limited in vivo tumour growth ability and spontaneous lung metastasis formation. This less aggressive phenotype is likely linked to reduced cell proliferation, assessed by a decreased staining of the proliferative marker PCNA in tumours, increased sensitivity to apoptosis, and strong inhibition in the EMT program observed in p140Cap expressing tumour cells.

To characterize the role of p140Cap in the NeuT preclinical model, we generated new Tg mice over-expressing p140Cap into the mammary gland under the MMTV promoter. MMTV-p140Cap Tg mice do not show any defects in the development or the differentiation of the mammary gland that could impair tumour growth (Supplementary Fig. 4). In the double Tg mice, expressing both NeuT and p140Cap, p140Cap expression reduces tumour burden, indicating that p140Cap is causative in limiting ERBB2 tumorigenic features in vivo. Indeed, p140Cap expression delays spontaneous tumour appearance and show decreased tumour masses, consistent with a decreased staining of the proliferative marker PCNA, with respect to NeuT mice. In addition, p140Cap expressing tumours show a different histology, reminiscent of that observed in the less aggressive human breast carcinoma[47,48]. When explanted, p140Cap Tg tumours do not display significant differences in apoptotic markers versus NeuT tumours. However, it is highly conceivable that the difference in total tumour burden reflects not only impaired tumour cell growth, but also the occurrence of local apoptotic events with remodelling of tumour structures, during tumour development. The in vivo analysis and the 3D Matrigel-Collagen cultures from primary cancer cells, suggest that p140Cap may limit the aggressiveness of ERBB2 tumours, both increasing tumour differentiation, restoring 'normal' mammary epithelial tissue morphogenesis[49,50] and differentially affecting the local tumour microenvironment[51]. In particular, upon apoptic stimuli and in 3D Matrigel-Collagen cultures, we observed that p140Cap cells have increased sensitivity to apoptosis. In the 3D conditions, p140Cap expression confers the ability to activate the apoptotic program and to give rise to internal lumen, typical of normal mammary epithelial cells during tissue morphogenesis[35].

The effect on EMT program is witnessed by the marked down-regulation of major EMT transcription factors, such as Snail, Slug and Zeb1 (ref. 36), accompanied by a reversion of the so-called 'cadherin switch' (that is, increase of the mesenchymal marker N-cadherin and a concomitant decrease of the epithelial marker E-cadherin), which is a canonical hallmark of EMT in cancer[15,36,52]. Indeed, p140 tumours display a homogeneous increased level of membrane E-cadherin, compared to NeuT tumours. Overall, the results point to the ability of p140Cap to counteract the EMT invasive program of ERBB2 tumour cells.

Notably, p140Cap expression significantly limits the ability of ERBB2 transformed cells to give rise to metastasis, both in experimental and in spontaneous metastasis assays. Indeed, when comparing tumours of the same size, p140Cap tumours give rise to a significantly lower number of spontaneous lung metastasis compared to NeuT tumours, suggesting that p140Cap affects metastatic spread. However, when analysing the metastatic lesions from p140Cap tumours, we observed a strong p140Cap expression only in smaller lung metastasis, rather than in larger ones, suggesting that p140Cap has also a strong effect on local metastatic growth. Therefore, from these data we can conclude that p140Cap significantly impairs metastasis acting both on tumour cell spreading and on metastatic growth, due to its ability to down-regulate tumour cell growth and to enhance apoptotic events.

Rac GTPase is a well-known mediator of human ERBB2 breast cancer progression[40–43], affecting signalling pathways impinging on tumour cell proliferation, apoptosis and acinar structure[53], as well as metastasis dissemination[54]. Here we show that p140Cap strongly impairs Rac activation in both human and mouse ERBB2 transformed cells. Indeed, in 3D Matrigel-Collagen morphogenetic assay, the Rac inhibitor NSC23766 (ref. 45) consistently decreased the area of the NeuT organotypic structures and restored cell polarity disrupted by the oncogene, thus recapitulating the effect of p140Cap expression. Of note, expression of a constitutively active form of Rac in p140Cap cells was able to rescue the aggressive ERBB2 phenotype, increasing acinar area and decreasing the percentage of polarized structures. These results further point to the mechanistic relevance of p140Cap/Rac counteraction as an essential step for limiting ERBB2 tumour progression. In the presence of p140Cap, only a constitutive alteration of Rac activation can reinstate the aggressive ERBB2 phenotype, suggesting that p140Cap may limit ERBB2 oncogenic features until at least significant Rac disregulation occurs.

Rac specific GEFs, like Dock, Tiam1 and PRex1, have also been shown to play a relevant role in breast cancers[42,54–56]. In particular, Tiam1 activation has been recently linked to the ERBB2 oncogene[57], where Tiam1-mediated Rac activation leads to uncontrolled actin dynamics that may compromise E-cadherin junctions, promoting metastasis[57–59]. Here working out the mechanisms underlying the observed decrease in Rac activation when p140Cap is expressed, we found a significant

**Figure 7 | p140Cap negatively controls ERBB2-driven migratory ability and Rac GTPase activity.** (**a,c,e,g**) Representative images of Transwell migration assays. $10^5$ cells were left to migrate for 24 h in the presence or the absence of 15% FBS, fixed, stained and counted. Histograms represent on the y axes the fold increase (ratio between the number of cells migrated in the presence and in the absence of FBS), from three independent experiments, performed in triplicate. Error bar: s.e.m. (**a**) p140Cap over-expressing (oe p140) or mock SKBR3 (mock) cells. (**c**) SKBR3 cells transiently transfected with ON-TARGET plus human *SRCIN1* small-interfering RNA (si p140) or ON-TARGET plus non-targeting siRNA (Dharmacon RNAi; si ctrl). This patented approach strongly prevents off-target effects. (**e**) MDA-MB-453 cells transiently transfected with ON-TARGET plus small-interfering RNA as in **c**. (**g**) Primary NeuT and p140 cancer cells. (**b,d,f,h**) Active Rac pull-down from cells like in (**a,c,e,g**). Eluted material (upper panels) and cell extracts (lower panels) run on 12% SDS–PAGE revealed with anti Rac antibodies. Histograms show the ratio between active and total Rac protein levels in arbitrary units (a.u.) from five independent experiments. Statistical significative differences were evaluated using umpaired t tests (*$P < 0.05$; **$P < 0.01$). Error bar: s.e.m. (**i**) Primary NeuT cells were grown in Matrigel/CollagenI 1:1 for 1 week, before seven days treatment with 80 µM Rac1 inhibitor NSC23766 and acini immunostained for GM130 (green), beta1 integrin (red) and DAPI for nuclei. Scale bar, 50 µm. Histograms represent quantification of acina area (left) and polarity (right) from three independent experiments. Differences in acina area were evaluated using a Mann–Whitney non parametric t-test (***$P < 0.001$). Error bar: s.e.m. (**j**) p140 primary cancer cells were infected with retroviral particles that express Rac1-V12 or empty vector (retro Ctrl). Cells were plated in Matrigel/Collagen I 1:1 and day 15 acinar structures were immunostained as in **i**. Scale bar, 50 µm. Quantification of acini area in a.u., percentage of polarized acina and percentage of acina with protrusions are reported. The values from two independent experiments are reported. Differences were evaluated using a Mann–Whitney non parametric t-test (***$P < 0.0001$; *$P < 0.05$). Error bar: s.e.m.

decrease in the activation of Tiam1 in p140Cap tumour cells. The observation that p140Cap associates in a molecular complex with Tiam1, suggests that this interaction reduces the activity of Tiam1 as a Rac GEF and that this could represent one major upstream event in negatively regulating Rac downstream pathways.

Data on the regulation of expression of p140Cap gene are currently limited. We show here that the p140Cap coding gene, *SRCIN1*, at chr17; 17q12-q2, is co-amplified in the *ERBB2* amplicon in almost 60% of ERBB2 amplified patients. *SRCIN1* amplification is caused by its proximity to the *ERBB2* gene, and correlates with p140Cap mRNA levels and with patient

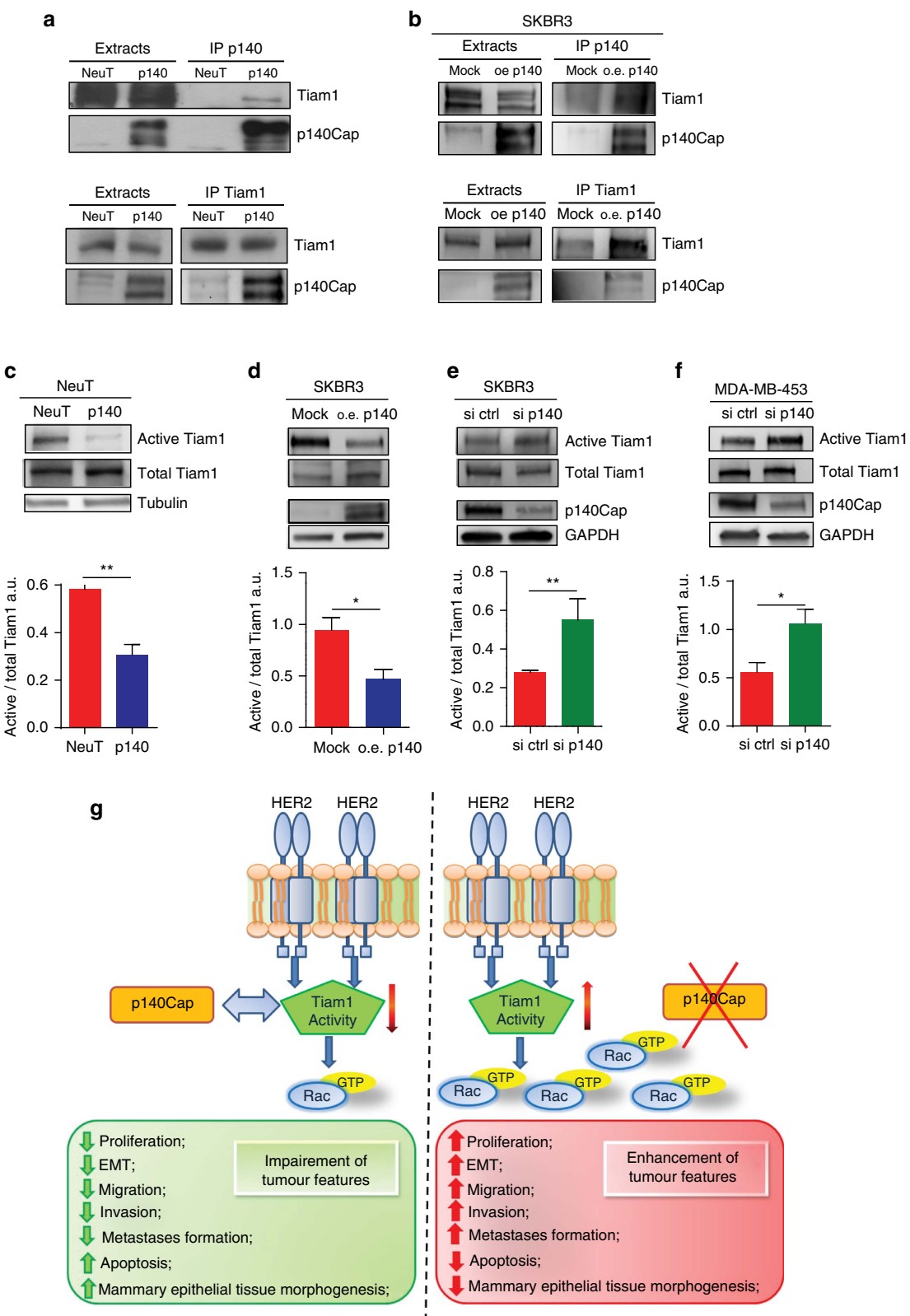

outcome. Interestingly, the aCGH data draw attention to a percentage of patients in which the *SRCIN1* gene is deleted (around 4–5% of the ERBB2 patients). These data highlight that in ERBB2 tumours, amplification of the *ERBB2* locus may lead to *SRCIN1* amplification or loss, thus contributing to the biological heterogeneity of this breast cancer subgroup[7–11]. However, besides amplification, additional mechanisms can account for alteration of p140Cap protein expression. Presently, data on the epigenetic regulation of p140Cap expression are not available. miR-150, miR-211, miR374a and miR346 have very recently been described as direct regulators of the p140Cap protein in lung, gastric and cutaneous squamous carcinoma cells[20–23], providing the first clues which link miRNAs to epithelial cancer cell features via the inhibition of p140Cap expression. Data on the ability of SRCIN1 to inhibit the osteosarcoma tumour cells proliferation have also been very recently reported[24].

In conclusion, our data are consistent with p140Cap exerting a suppressive function on ERBB2 oncogenic features and with it having a regulatory impact on molecular pathways that ERBB2 exploits for tumour progression (Fig. 8g). Moreover, p140Cap expression is advantageous for patient survival, strongly suggesting that p140Cap is still causal in limiting ERBB2 tumour aggressiveness within the complexity of the *ERBB2* amplicon. Indeed, our data provide the first evidence, to our knowledge, that a gene in the *ERBB2* amplicon may counteract ERBB2 oncogenic properties in breast cancer. Altogether, these data highlight the potential clinical impact of p140Cap expression and of p140Cap-regulated pathways in human ERBB2 breast tumours as new therapeutic targets.

## Methods

**Antibodies and cell lines.** Mouse monoclonal antibodies to p140Cap were produced at the Antibody production facility of the Dept of Molecular Biotechnology and Health Sciences, University of Torino. A recombinant p140Cap protein, obtained in Escherichia coli by fusing the sequence corresponding to amino acids 800–1,000 of mouse *SRCIN1* gene to the Glutathione S-transferase (GST) was incubated with 4% paraformaldehyde in $1 \times$ PBS—pH 7.4 for 30 min, at a concentration of 750 µg ml$^{-1}$, dialysed, and injected into p140Cap KO mice[60] for enhancing immunogenic activity. The resulting purified monoclonal antibodies were characterized by western blotting and IHC as shown in Supplementary Fig. 1. For western blot analysis, the following antibodies were used: anti Snail (#3895, 1:1,000), anti Caspase-3 (9,665, 1: 1,000), anti phospho Paxillin (Tyr118; #2541, 1:1,000), anti Paxillin (#2542, 1:1,000), anti phospho p130Cas (Tyr410; #4011. 1:1,000) and anti phospho-Src (Tyr416; #2101, 1:1,000; Cell Signaling, Beverly, MA), anti N-cadherin (ab10203, 1:1,000) and anti GFP (ab13970, 1:500; Abcam, Cambridge, UK), anti c-ErbB2/c-Neu (Ab-3, OPL15, 1:1,000; Calbiochem, Merck KGaA, Darmstadt, Germany), anti Rac1 (#05–389 clone 23A8, 1:2,000), anti GAPDH (MAB374, 1:8,000) and anti p1248Y ERBB2 (#06–229, 1:1,000; Millipore, Billerica, MA, USA), anti beta1 Integrin CD29-PE (12-0299-41, 1:200; eBioscience, San Diego, CA, USA), anti GM130 (6,10,823, 1:300), anti p130Cas (6,10,272, 1:2,500) and E-Cadherin (6,10,182, 1:2,500; BD Transduction Laboratories, Franklin Lakes, NY), anti Tiam1 (C-16, 1:1,000), anti Actin (I-19, 1:1,000), anti Src (B-12, 1:1,000), and anti Cyclin D1 (H-295, 1:1,000; Santa Cruz Biotechnologies, Palo Alto, CA, USA), and anti

Tubulin (T5168, 1:8,000; Sigma-Aldrich Co, Italy). Secondary antibodies conjugated with peroxidase were purchased from GE Healthcare. Alexa Fluor Dye secondary antibodies were obtained from Invitrogen (Carlsbad, CA, USA). For immunohistochemistry, slides were stained with the following primary antibodies: rabbit polyclonal anti-HER2 (A0485, 1:700, Dako, Carpinteria, CA, USA), mouse monoclonal anti-PCNA (M0879, 1:800, Dako, Carpinteria, CA, USA), rabbit polyclonal anti-Caspase3 (af835, 1:350, R&D System, Minneapolis,MN, USA), rat monoclonal anti-CD31 (5,50,274, 1:40, BD Pharmingen, San Jose, CA, USA) mixed with rat monoclonal anti-CD105 (5,50,546, 1:40, BD Pharmingen, San Jose, CA, USA), rabbit polyclonal anti-Keratin 5 (PRB-160 P, 1:2,000, Covance, USA), guinea pig polyclonal anti-Keratins 8/18 (GP11, 1:750, PROGEN Biotechnik GmbH, Heidelberg, Germany) and mouse monoclonal antibody anti-p140Cap (1:500, see above) followed by the appropriate secondary antibodies. Immunoreactive antigens were detected using streptavidin peroxidase (Thermo Scientific UK) and the DAB Chromogen System (Dako, Carpinteria, CA, USA) or alkaline phosphatase conjugated streptavidin (Thermo Scientific UK) and Vulcan fast red chromogen (Biocare Medical, Concord, CA, USA). For immunofluorescence, slides were stained with the mouse anti-human E-cadherin antibody (M3612, 1:50, Dako, Carpinteria, CA, USA) followed by secondary antibody conjugated with Alexa 488 (A11029, 1:200, Invitrogen, Life Technologies, Monza, Italy). Lapatinib was bought from Selleckchem (Munich, Germany). Rac inhibitor (NSC23766) was bought from Calbiochem (Merck KGaA, Darmstadt, Germany. Glutathione-Sepharose, Protein G-Sepharose, PVDF, and films were obtained from GE Healthcare (Buckinghamshire, UK). Culture media were from Invitrogen (Carlsbad, CA, USA). Fetal Calf serum (FCS) was from EuroClone (Pero, Milano, Italy). SKBR3, MDA-MB-453 and MDA-MB-231 cells were obtained from ATCC (LGC Standards S.r.l.—Italy Office, Italy). SKBR3 cells were cultured in McCoy's 5a medium, supplemented with 15% FCS. MDA-MB-453 cells were cultured in DMEM 10% FCS. MDA-MB-231 cells were cultured in DMEM 10% FCS. NeuT-TUBO cells were derived from a spontaneous breast tumour arisen in a female BALB/c-MMTV-NeuT mice[37] and cultured in DMEM 20% FCS.

**Human breast cancer immunohistochemical analysis.** IHC analysis of p140Cap expression was performed on formalin-fixed paraffin-embedded tissue microarrays, prepared with tumour breast specimens, using a mouse monoclonal antibody anti-p140Cap (Supplementary Fig. 1), which was used at a dilution of 1:1,000 following an antigen retrieval procedure in EDTA pH 8.0. Immunocomplexes were visualized by the EnVision + HRP Mouse (DAB + ) kit, DAKO (K4007), and acquired with the Aperio ScanScope system (Leica Biosystems). Informed consent was obtained from all subjects. For the purpose of correlation with clinical and pathological parameters, tumours were classified based on the intensity of p140Cap staining as p140Cap-Low (IHC score <1) and p140Cap-High (IHC score ≥1).

**aCGH and gene expression analyses.** Normalized aCGH profiles from 200 ErbB2 breast cancers together with matching gene expression profiles from 50 cases were obtained from the data described in ref. 8. Correlation analyses between gene CN, determined by aCGH, and mRNA expression for *SCRIN1* were performed using the Pearson correlation as described in refs 8,61.

**FISH analysis of *SRCIN1* gene status.** A specific SRCIN1 locus probe was prepared from the BAC RP11-606B22 (17q12) clone, obtained from BAC PAC Resources Center (Children's Hospital, Oakland Research Institute, USA). The BAC was directly labelled with red SpectrumAqua-dUTP (Abbott Molecular, Europe), using the BioPrime DNA Labeling System (Invitrogen Corporation, USA) according to manufacturer's instructions. An alpha satellite probe specific for chromosome 17 (CEP17; Abbott Molecular) directly labelled with green fluorochrome, was used as a control probe. To further analyse the position and strength of the signal, the presence/absence of background, cross-hybridization and, finally, the hybridization efficiency, the BAC clone was tested on metaphase and interphase healthy donor cells obtained using conventional cytogenetic

**Figure 8 | p140Cap expression negatively regulates Tiam1 activity.** (**a,b**) Extracts from NeuT and p140 expressing cancer cells, and p140 overexpressing (o.e.), or mock (mock) SKBR3 cells were immunoprecipitated with antibodies to p140Cap (upper panels) or Tiam1 (lower panels). Cell extracts and immunoprecipates were run on 6% SDS–PAGE and blotted with antibodies to p140Cap and Tiam1. Representative images from five independent experiments are shown. (**c–f**) The level of active Tiam1 was determined using the active Rac-GEF assay kit in NeuT, p140 primary cancer cells, p140 o.e. or mock (mock) SKBR3 cells, and p140 silenced SKBR3 (si p140) and MDA-MB-453 (si p140) cells. Equal amount of extracts were incubated for 1 h at 4 °C with Rac G15A agarose beads. Active Tiam1 and total Tiam1 levels were determined using an anti-Tiam1 antibody for western blot detection, from eluted material and input fractions, respectively. Antibodies to tubulin and GAPDH were used as loading controls. The histogram represents the quantification of active Tiam1 in three independent experiments, normalizing active Tiam1 levels to the corresponding total Tiam1 levels in arbitrary units (A.U.). In **c–f**, statistical significative differences were evaluated using unpaired $t$-tests (*$P<0.05$; **$P<0.01$). Error bar: s.e.m. (**g**) p140Cap exhibits a suppressive function on ERBB2 tumour features. In ERBB2 cancer cells, when p140Cap is expressed, proliferation, EMT, migration and metastasis formation are impaired and cancer cells enhance apoptosis and restore the proper mammary epithelial tissue morphogenesis disrupted by the ERBB2 oncogene. Moreover, the Tiam1/Rac signalling pathway is strongly decreased, through the ability of p140Cap to associating with Tiam1 and to downregulating its activity. On the contrary, when p140Cap is undetectable, Tiam1/Rac signalling pathway is active, and cancer cells exhibit an aggressive phenotype. The molecular mechanisms here reported link p140Cap expression with decreased metastatic risk in ERBB2 patients.

methods. The PathVysion *ERBB2* DNA probe kit was used (Abbott Molecular, Europe) for the *ERBB2* locus. FISH with the two probes mix, *SRCIN1*/CEP17 and *ERBB2*/CEP17, was routinely performed on formalin-fixed paraffin-embedded tissue. Red (*SRCIN1*) and green (*CEP7*) spots on significant selected areas were automatically acquired, using Metafer, by a MetaSystem scanning station (Carl Zeiss MetaSystems Gbmh), equipped with an AxioImager epifluorescence microscope. The first automatic lecture of the slides, made using the PathVysionV2 software, was performed on the acquired images with Isis software (Zeiss). The ASCO/CAP 2013 Guideline Recommendations for *ERBB2* Testing in the Breast were used for the interpretation of both FISH probes: positive for amplification with *ERBB2*—*SRCIN1*/CEP17 ratio $>2.0$ or with average *ERBB2*—*SRCIN1* CN $>6$; negative for amplification with *ERBB2*—*SRCIN1*/CEP17 ratio $<2.0$ or *ERBB2*—*SRCIN1* copy $<4$. Gene loss was considered to occur when an average *ERBB2*—*SRCIN1* CN $<1.8$ was found and gene gain when CN was $>3<6$. Finally, when heterogeneity was present (such as the presence in the same sample of amplified and not amplified cells), we considered samples where the amplified cell population consisted of $>10\%$ tumour cells as being amplified.

### Generation of the MMTV-p140Cap transgenic mice.
Full-length mouse p140Cap cDNA was inserted into pspT2 MMTV-LTR plasmid and microinjected at 3.4 ng/microliter in the pronucleus of fertilized eggs from FVB/NJ mice (Charles River, Calco, Italy) according to standard protocols[62]. Transgene integration was tested via PCR analysis of genomic DNA, with the primers: 5′-TGGCCCTGCGAGGTCAGCAGGACA-3′, 5′-ATCCTGCTGAAGC-CCAG GGGCAGC-3′. Heterozygous mice carrying the mutated rat HER-2/neu oncogene driven by the MMTV-LTR promoter (MMTV-NeuT mice), either on FVB/NJ (FVB-MMTV-NeuT) or BALB/c (BALB/c-MMTV-NeuT) background, are well-characterized transgenic models of spontaneous NeuT mammary adenocarcinoma[27–29,63]. p140Cap/Neu-T double transgenic mice were generated by crossing MMTV-p140Cap transgenic female (FVB background) with either FVB-MMTV-NeuT or BALB/c-MMTV-NeuT. The progeny was screened for both the transgene by PCR. The mice that were positive for both transgenes were included in further analyses, while animals positive only for the NeuT transgene were used as controls ($n = 12$ for each group). The size of the tumours was evaluated weekly using calipers in blind experiments. The project had been approved by the Internal Bioethical Committee of the Department of Molecular Biotechnology and Health Sciences of the University of Torino. The handling of mice in our animal house meets the requirements of Italian law (authorization D.M. no. 279/95B 27/11/1995 and Ministry of Health 49/2014-PR to PD) and follows the dispositions of 'D.L. no. 116, 27/1/1992 in relation to animal use and protection in scientific research'.

### Immunohistochemistry and immunofluorescence analyses of NeuT tumours.
Tumour samples were fixed in 10% neutral buffered formalin and embedded into paraffin or fixed in 4% PFA and frozen in a cryo-embedding medium (OCT, BioOptica); 5 μm slides were cut and stained with Hematoxylin (BioOptica) and Eosin (BioOptica) for histological examination. The percentage of PCNA or Caspase3-positive cells was evaluated on digital images of 3 tumours per group ($4–6 \times 200$ microscopic fields per sample); clear brown nuclei were regarded as positive cells and the percentage of labelling index (number of positive cells/total cells $\times 100$) was calculated for each field, by two pathologists, independently, and in a blind fashion. The vascularization was analysed evaluating CD31-105+ endothelial cells on digital images of 3 tumours per group ($6 \times 200$ microscopic fields per sample) with Adobe Photoshop by selecting red stained vessels with the Magic Wand Tool and reporting the number of pixels indicated in the histogram window. For both experimental and spontaneous lung metastasis assay, lungs were fixed in 10% neutral buffered formalin and paraffin-embedded. To optimize the microscopic metastases and ensure systematic uniform and random sampling, lungs were cut transversally, to the trachea, into 2 mm thick parallel slabs with a random position of the first cut in first 2 mm of the lung, resulting in 5–8 slabs for lung. The slabs were embedded cut surface down and sections were stained with Hematoxylin and Eosin (BioOptica, Milan, Italy). The metastatic lung tissue was evaluated with Adobe Photoshop by selecting metastases with the lasso tool and reporting the number of pixels indicated in the histogram window as percentage of the total lung area. For immunofluorescence, slides were stained with the mouse anti-human E-cadherin antibody followed by secondary antibody conjugated with Alexa 488. Image acquisition was performed using Zeiss LSM 510 META confocal microscope. The E-cadherin mean fluorescence intensity was evaluated on the digital images of three tumours per group ($4 \times 200$ microscopic fields per sample) with ImageJ, using the Mean Grey Value: for RGB images, the mean was calculated by converting each pixel to grayscale using the formula grey $= 0.30$red $+ 0.59$green $+ 0.11$blue if 'Unweighted RGB to Grayscale Conversion' was checked in Edit > Options > Conversions. Whole-mount preparation were performed as described in ref. 60. The fourth abdominal mammary glands were analysed from at least three mice for age group. Only whole mounts that contained the entire ductal network including the primary duct and were free of mounting artifacts such as tissue folds were used for subsequent image analysis. A digital photomicrograph was taken of each whole-mount using a Leica MZ6 stereo microscope fitted with a Nikon Coolpix colour digital microscope camera. Within each age group, a consistent

magnification was established that allowed the entire epithelial complex to be captured in a single image. For each age group, the photomicrographic settings remained constant. Four different measurements were obtained from each whole-mount image using Photoshop software. TEB count was performed only on 6 weeks of age glands. Ductal length (pixels) was measured by drawing and measuring a straight line caliper from the most distal point of the ductal network to the nipple. Ductal network area tumours from NeuT mice and xenografts were routinely fixed in 10% formaldehyde buffer (pH 7.4) for 24 h, paraffin-embedded and processed for immunohistochemical analysis with standard procedures[64].

### Isolation of primary cancer epithelial cells from mammary gland tumours.
Cells from tumours were isolated as described in refs 29,62. Briefly, tumours were surgically excised from 17-week-old BALB/c Neu-T and p140-NeuT mice and finely chopped. Tumour cell aggregates were then incubated in trypsin (0.25% in EDTA) for 2 h at 37 °C, washed in DMEM, centrifuged at low speed and then plated in 20% FBS/DMEM. After the sprouting of cells from tissue fragments, the cultures were periodically and briefly washed (1–3 min) with trypsin-EDTA to detach contaminating fibroblasts without damage to epithelial areas. Two months after plating, established epithelial cell populations were selected by several subculturing steps.

### Three-dimensional cultures of primary cancer cells.
For 3D-Matrigel cultures, eight-well Chamber slides (Corning) and Growth-factor-reduced Matrigel (BD Transduction Laboratories) were used. Three-dimensional culture assays were performed in agreement with protocols reported in: http://muthuswamy-lab.cshl.edu/protocols. Briefly, NeuT, p140-NeuT primary cancer cells or NeuT and p140-NeuT stable infected cells were embedded as single cells in Matrigel/Collagen I 1:1 and left to grow for 15 days. When indicated, the Rac inhibitor (NSC23766) was added to culture medium for the last 7 days. After 15 days, acini were subjected to immunostaining as described in http://muthuswamylab.cshl.edu/protocols/IF protocol.pdf. Images were taken using a Zeiss microscopy (Oberkochen, Germany) equipped with an Apotome module at $\times 40$ magnitude. For immunoblotting analysis, NeuT or p140 cells were released from Matrigel-Collagen gels using BD cell recovery solution (BD Biosciences) and protein were extracted with RIPA buffer (50 mM Tris (pH7.5), 150 mM NaCl, 1% Triton X100, 1% Na Deoxycolate, 0.1% SDS and protease inhibitors). Cell lysates were centrifuged at 13,000 g for 15 min and the supernatants were collected and assayed for protein concentration using the Bio-Rad protein assay method (Biorad, Hercules, CA, USA). Proteins were run on SDS–PAGE under reducing conditions.

### Retrovirus production and cell infection.
To over-express p140Cap into SKBR3, NeuT-TUBO and MDA-MB-231 cells, p140Cap cDNA was cloned into pBabe-puro. The plasmid that encodes GFP-Rac1V12 was purchased from Addgene (Cambridge, MA, USA). The retroviruses particles were produced by the calcium phosphate transfection of Platinum Retroviral Packaging Cell Lines (Cell BioLabs), in 10 cm dishes. 48 h after transfection, supernatant that contained the retrovirus particles was collected, filtered through a 45 μm syringe filter and added directly to subconfluent cells. After 48 h, cells were washed and cultured with a selection medium containing puromycin (Sigma) at a final concentration of 1 μg ml$^{-1}$. The efficiency of infection was controlled by western blot analysis. For SKBR3, NeuT-TUBO and MDA-MB-231 cells, individual clones were isolated 20 days after the start of the selection. Four individual positive clones were pooled together to rule out clonal artifacts.

### *In vivo* tumour growth and experimental and spontaneous metastasis assay in NeuT cells.
Five-week-old female CD-1 Nude Mouse were purchased from Charles River Laboratories (Calco, Italy) and treated in accordance with the European Community guidelines. $1 \times 10^6$ NeuT or p140 cells were mixed with 150 μl DMEM and then injected subcutaneously into the left and right inguinal region of female nude mice. The size of the tumours was evaluated weekly using calipers in blind experiments. For experimental lung metastasis assay, NeuT-TUBO and p140-TUBO cells were trypsinized, resuspended in PBS, and then $5 \times 10^4$ cells (in 0.1 ml) were injected via the lateral tail vein of 7-week-old female NSG mice (NOD.Cg-Prkdcscid Il2rgtm1Wjl/SzJ) from Charles River Laboratories (Calco, Italy; $n = 5$ for group for each experiment). Mice were killed 25 days after injection and lungs were fixed in 10% neutral buffered formalin (BioOptica) and paraffin-embedded. To optimize the detection of microscopic metastases and ensure systematic uniform and random sampling, lungs were cut transversally, to the trachea, into 2 mm thick parallel slabs with a random position of the first cut in first 2 mm of the lung, resulting in 5–8 slabs for lung. The slabs were then embedded cut surface down and sections were stained with Hematoxylin and Eosin (BioOptica).The metastatic lung tissue was evaluated with Adobe Photoshop by selecting metastases with the lasso tool and reporting the number of pixels indicated in the histogram window as percentage of the total lung area. For spontaneous lung metastasis assay, NeuT-TUBO and p140-TUBO cells were trypsinized, resuspended in PBS, and then $10^5$ cells (in 0.1 ml) were injected into the right fat pad of 7-week-old female NSG mice ($n = 5$ for group for each experiment). We monitored mammary tumour growth by regular measurements using a digital caliper. Tumours were surgically removed when reached

a 10 mm diameter. After 5 weeks, mice were killed and lungs were explanted and processed as previously described.

**Analysis of EMT markers by qRT–PCR.** Total RNA was extracted using the RNeasy Mini kit (Qiagen, CA) with DNase I treatment and quality controlled by electrophoresis on 0.8% agarose gel. RT–PCR was performed on 0.5–1 µg total RNA with the SuperScript ViloTM cDNA Synthesis kit from Invitrogen. Gene expression was assessed by quantitative real-time PCR with the GeneAmp 7,500 system and Taqman chemistry (Applied Biosystems, CA). Each sample was tested in triplicate. The Δ-Ct method was used to calculate relative fold-changes normalized against three different housekeeping genes. Taqman Gene Expression Assay IDs (Applied Biosystems, CA) were: Mm00441533-g1 (Snail, snail1, NM_011427.2), Mm00441531-m1 (Slug, snail2, NM_011415.2), Mm00486906-m1 (E-cadh, cdh1, NM_009864.2), Mm00483213-m1 (N-cadh, cdh2, NM_007664.4), Mm00495564-m1 (zeb1, NM_011546.2), Mm99999915-g1 (GAPDH), Mm01197698-m1 (Gusb, NM_010368.1), Mm00607939-s1 (Actb, NM_007393.3).

**Immunoprecipitation and immunoblotting.** Cells were extracted using a RIPA buffer (see above). Cell lysates were centrifuged at 13,000 g for 15 min and the supernatants were collected and assayed for protein concentration using the Bio-Rad protein assay method (Biorad, Hercules, CA, USA). Proteins were run on SDS–PAGE under reducing conditions. For co-immunoprecipitation experiments, 1 mg of proteins was immunoprecipitated with antibodies to p140Cap for 2 h at 4 °C in the presence of 50 µl protein G-Sepharose beads. Following SDS–PAGE, proteins were transferred to PVDF membranes, incubated with specific antibodies and then detected with peroxidase-conjugated secondary antibodies and the chemiluminescent ECL reagent. When appropriate, the PVDF membranes were stripped according to manufacturers' recommendations and re-probed.

**Transient silencing of p140Cap in SKBR3 and MDA-MB-453.** Transient transfections of ON-TARGET plus human SRCIN1 small-interfering RNA (siRNA) or ON-TARGET plus non-targeting siRNA (Dharmacon RNAi, GE Healtcare, Buckinghamshire, UK) were performed with Lipofectamine 2,000 (Invitrogen, USA) according to manufacturer's protocol. This patented approach is the best strategy to prevent off-target effects caused by both the sense and antisense strands while maintaining high silencing potency. Briefly, cells were plated on six-well plate and transfected at 80% confluency. Either 5 µl of 20 microMolar p140Cap siRNA or non-targeting siRNA were added to each well, and cells were incubated for 48 h at 37 °C in a humidified $CO_2$ incubator. Transfected cells were used for different assay.

**Proliferation and apoptosis assays.** To assess the NeuT and p140 cell proliferation rate, $15 \times 10^4$ cells were seeded per well in a 24-well plate and counted at the indicated times. Quantification of Neu-T TUBO cell growth was done by MTT assay. For apoptosis assays, NeuT and p140 cells were serum-starved for 12 h or detached and kept in suspension for 12 h. Apoptosis was assayed by annexin V staining (BD Biosciences, San José, CA, USA) or by immunoblotting with anti caspase-3 antibody (Cell Signaling).

**Transwell migration assay.** For the migration assay, Transwell chambers (Corning, Corning, NY, USA) were coated with 10 µg ml$^{-1}$ type I collagen (Corning). Cells were detached using 5 mM EDTA and suspended in serum-free medium. The cells were seeded on top of the 8.0 µm pore size at a density of $1 \times 10^5$ cells per well in 100 microliters of serum-free medium 0.1% BSA. As chemoattractant, 700 µl of medium containing 15% FBS was placed in the lower chamber. After 24 h, the cells on the top surface of the filter were removed with a cotton swab, and the migrating cells on the lower surface of the membrane filter were fixed and stained with Diff-Quick kit (Medion Diagnostics International Inc, Miami. USA), and counted using a light microscope $\times 10$ magnification.

**Rac GTPases *in vitro* activity assay.** Cells were washed twice on ice with PBS and then lysed in a MLB buffer (25 mM EDTA, 150 mM NaCl, 2% glycerol, 1% NP40, 1 mM EDTA, 10 mM MgCl2, 10 µg ml$^{-1}$ each of leupeptin, pepstatin and aprotinin). For pull-down experiments glutathione-coupled Sepharose 4B beads bound to recombinant GST-PAK CRIB domain fusion proteins were incubated with cell extracts at 4 °C for 30 min, eluted in Laemmli buffer and analysed for the presence of Rac1 by western blot.

**Tiam1 activity assay.** Assays were performed using the active Rac-GEF assay kit (Cell Biolabs, San Diego, USA) according to the manufacturer's instructions. Briefly, cells were washed twice with ice-cold PBS and lysed in ice-cold 1× Assay/Lysis Buffer (1 mM PMSF, 10 µg ml$^{-1}$ leupeptin and 10 µg ml$^{-1}$ aprotinin). Extracts were incubated with 40 µl of resuspend Rac1 G15A agarose bead slurry and incubate for 1 h at 4 °C. Beads were washed three times with the 1× Assay/Lysis Buffer, resuspended in 40 µl of 2× reducing SDS–PAGE sample buffer and boiled for 5 min. Pull-down supernatant were subjected to SDS–PAGE electrophoresis and western blotting with anti-Tiam1 antibody.

**Statistical analysis.** Tissue microarray data analysis was performed using JMP 10.0 statistical software (SAS Institute, Inc). The association between p140Cap expression and clinico-pathological parameters was evaluated using the Pearson chi-square test. For univariate and multivariate analysis, hazard ratios and 95% confidence intervals were obtained from the Cox proportional regression method. Differences in the growth rate of mouse tumours were analysed with Fisher's Exact Test, or two-way ANOVA followed by Bonferroni multiple comparison *post hoc* test. Differences in acina area were evaluated using a Mann–Whitney non parametric *t*-test. For quantification, statistical significative differences were evaluated using unpaired *t*-tests. Error bar: s.e.m. using the Student's *t*-test.

**Data availability.** All other remaining data are available within the Article and Supplementary Files, or available from the authors upon request.

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

## Acknowledgements

We thank: M.F. Brizzi, and D. Noonan for critical reading of the manuscript and insightful discussion; C. Luise, G. Jodice, D. Ricca, M. Coazzoli and the Molecular Pathology of the Molecular Medicine Program at IEO for technical support. This work was supported by the Associazione Italiana Ricerca Cancro (AIRC; IG-15,399 to PD; IG-11,346 to S.C.); Ministero Università Ricerca (MIUR; PRIN 2015 to P.D. and to P.P.D.F.); Regione Piemonte (Project Acronym: Oncoprot: Onco-proteins, Druidi: Drug Innovation and Discovery to P.D.); Compagnia San Paolo, Torino; Progetto d'Ateneo, Università di Torino 2011 to P.D. and E.T.

## Author contributions

S.G., J.C., V.S., S.A., S.C., D.T., G.C. performed cell biology experiments; S.G., S.L., I.R., N.S., A.A., F.C., M.I., L.S., V.S., N.V. and A.L. generated and characterized the animal models; J.S., L.V.C., I.C., A.S., K.D.; A.A. G.B., S.P., S.C., P.P.D.F., E.M. and P.P. conceived and generated the data on human cohorts; P.D.S., E.T. and P.D. conceived the study; S.G., S.P., P.P.D.F, E.T. and P.D. wrote the paper with input from all authors.

## Additional information

**Competing interests:** The authors declare no competing financial interests.

DOI: 10.1038/ncomms16203    OPEN

# Author Correction: The scaffold protein p140Cap limits ERBB2-mediated breast cancer progression interfering with Rac GTPase-controlled circuitries

Silvia Grasso, Jennifer Chapelle, Vincenzo Salemme, Simona Aramu, Isabella Russo, Nicoletta Vitale, Ludovica Verdun di Cantogno, Katiuscia Dallaglio, Isabella Castellano, Augusto Amici, Giorgia Centonze, Nanaocha Sharma, Serena Lunardi, Sara Cabodi, Federica Cavallo, Alessia Lamolinara, Lorenzo Stramucci, Enrico Moiso, Paolo Provero, Adriana Albini, Anna Sapino, Johan Staaf, Pier Paolo Di Fiore, Giovanni Bertalot, Salvatore Pece, Daniela Tosoni, Stefano Confalonieri, Manuela Iezzi, Paola Di Stefano, Emilia Turco & Paola Defilippi

*Nature Communications* **8**:14797 doi: 10.1038/ncomms14797 (2017); Published online 16 Mar 2017; Updated 30 Mar 2018

In the original version of this Article, the affiliation details for Anna Sapino were incorrectly given as Department of Medical Sciences, University of Torino, 10126 Torino, Italy instead of Candiolo Cancer Institute-FPO, IRCCS, Str. Prov. 142, km 3.95, I-10060, Candiolo (To), Italy. This has now been corrected in both the PDF and HTML versions of the Article.

