## [Peer Review File · Nature Communications]

Reviewers' comments:

Reviewer #1 (Remarks to the Author): Expert on Rac and cancer migration

The study by Grasso et al. focuses on the role of p140CAP in ERBB2-mediated breast cancer progression. The same group has previously contributed much of what is known about p140CAP, including its binding to p130CAS and CSK, its regulation of SRC activity, and its correlation with a less aggressive breast cancer disease. Studies presented in this manuscript are an extension of the prior work and show the common co-amplification of SRCIN1 (the gene for 140CAP) with ERBB2 in breast cancer patients, as well as its correlation with a less aggressive disease specifically in ERBB2-amplified tumors. Transgenic mice and cell lines are used to prove the ability of co-expressed p140CAP to suppress ERBB2-induced phenotypes, including induction of EMT, loss of cell polarity, cell migration, experimental metastasis and cancer progression. The effects of p140CAP are related to Rac1 activation by the RacGEF TIAM1, which is shown to associate with p140CAP. In general, this is a very well controlled study with interesting findings that relate to the progression of HER2 positive breast cancer.

One major issue is the almost exclusive focus on migration processes. From both the mice experiments and the cell line data it is obvious that there are effects of p140CAP expression on cell growth. This may be related to effects on proliferation or apoptosis. Both of these processes have been linked to HER2 signaling, as well as to CSK, SRC and p130CAS, including effects on anchorage-independent transformed cell growth. Furthermore, Rac1 activity has been linked previously to loss of acini lumen formation through effects on cellular apoptosis. Similar concerns also exist relating to the experimental tail-vein metastasis assay, which often reflects changes in anchorage-independent growth or apoptosis.

There is lack of any experimentation and/or discussion of the impact of either CSK/SRC activation, or p130CAS in p140CAP effects towards ERBB2 signaling. SRC activation is critical for the transformed phenotype of activated ERBB2 and promotes Rac1 activation. Both p130CAS and TIAM1 are SRC substrates that eventually regulate Rac1 activity. This is also true for VAV2 another important Rac1 inducer in breast cancer cells. Given the prior findings of this group related to the role of p140CAP in SRC activation it is unclear why these relationships were not probed experimentally any further.

As it stands, it is unclear whether this study relates to prior roles of p140CAP reported by the same group or not. While one may argue that novelty is somewhat reduced by further clarifying an already suggested mechanism of action, this reviewer believes this would add strength and not weakness to this story and help clarify the mechanism of p140CAP action.

Reviewer #2 (Remarks to the Author): Expert on breast cancer mouse models

This manuscript deals with the observation that in 60% of HER2 amplified breast cancers, a p140Cap scaffold protein encoded by the SRC1N1 is also co-amplified. The authors provide evidence that amplification of SRC1N1 is correlated with good outcome. The authors show that overexpression of SRC1N1 in the mammary epithelium can reduce tumor burden in a MMTV-NeuT model. Mechanistically the authors argue that p140Cap suppresses tumor growth by interfering with TIAM activation of Rac 1. Although this is an interesting observation, the quality of data and the depth of analyses presented in this manuscript limit my enthusiasm for the paper.

Specific Points;

1. The FISH analyses shown in Fig, 2C is very poor quality (either the image or primary data) and thus is impossible to interpret

2. The characterization of the MMTV/ SRC1N1 strains is very cursory-There is no data shown to show the range of tissues the transgene is expressed. There is no immunohistochemical analyses demonstrating that SRC1N1 is expressed in mammary epithelial manner.
3. Does ectopic expression of SRC1N1 impair normal mammary gland differentiation or development? This has direct bearing as the effect on mammary tumor induction indirect.
4. The quality of data showing apoptotic and proliferative capacity (Fig, 3E) is very poor. There is quantitative assessment of apoptotic and proliferative status-The authors should also examine whether tumor impairment reflects the level of angiogenic infiltration (CD31 staining)
5. Again the quality of data shown in Figure 4H is poor.
6. The authors should stain the tumor sections with CK8, CK14 and CK5 to establish whether the luminal status of the tumors is altered.
7. The effects of ectopic expression of SRC1N1 on the spontaneous metastasis should be included in the analyses.
8. Figure 5 should include immunohistochemical analyses of E-cadherin staining in addition to immunoblot
9. Figure 5E is of very poor quality and should be removed or repeated
10. Figure 6G and H are of very poor quality (Indeed only small selected fields are shown)
11. The authors should inactivate SRC1N1 in a number of ErbB2 amplified breast cancer lines rather than the single SKBr3 line shown.

Minor Comments

The reference to Neu T strain used is not a primary reference. Because several strains of this type were generated that may have different properties this is important point

Throughout the manuscript blots comprise 2 or 3 samples at most. An inclusion of at least 5 samples for each experimental point would increase the confidence in the validity of the data

Many of immunofluorescence images shown are not of publication quality.

Rebuttal letter to REVIEWERS' COMMENTS:

Reviewer 1 (Remarks to the Author): Expert on Rac and cancer migration

The study by Grasso et al. focuses on the role of p140CAP in ERBB2-mediated breast cancer progression. The same group has previously contributed much of what is known about p140CAP, including its binding to p130CAS and CSK, its regulation of SRC activity, and its correlation with a less aggressive breast cancer disease. Studies presented in this manuscript are an extension of the prior work and show the common co-amplification of SRCIN1 (the gene for 140CAP) with ERBB2 in breast cancer patients, as well as its correlation with a less aggressive disease specifically in ERBB2-amplified tumors. Transgenic mice and cell lines are used to prove the ability of co-expressed p140CAP to suppress ERBB2-induced phenotypes, including induction of EMT, loss of cell polarity, cell migration, experimental metastasis and cancer progression. The effects of p140CAP are related to Rac1 activation by the RacGEF TIAM1, which is shown to associate with p140CAP. In general, this is a very well controlled study with interesting findings that relate to the progression of HER2 positive breast cancer.

We thank the reviewer for his/her kind comments.

One major issue is the almost exclusive focus on migration processes. From both the mice experiments and the cell line data it is obvious that there are effects of p140CAP expression on cell growth. This may be related to effects on proliferation or apoptosis.

Both of these processes have been linked to HER2 signaling, as well as to CSK, SRC and p130CAS, including effects on anchorage-independent transformed cell growth. Furthermore, Rac1 activity has been linked previously to loss of acini lumen formation through effects on cellular apoptosis. Similar concerns also exist relating to the experimental tail-vein metastasis assay, which often reflects changes in anchorage-independent growth or apoptosis.

As highlighted by Reviewer 1, data obtained from in vivo and in vitro experiments suggest that the expression of p140Cap may have a role in controlling cell growth. To investigate whether the presence of p140Cap correlates with a reduction in cell growth we performed cell proliferation in standard culture conditions for NeuT/p140 cells. As shown in Supplementary Figure 6A (also included here as Figure 1A for referee 1), NeuT and p140 cells proliferate at the same rate. This is consistent with the comparable levels of cyclin D1 observed by western blot analysis (Figure 1B for referee 1). In an anchorage-independent assay, after 4 weeks of soft agar growth, both NeuT and p140 cells formed sporadic colonies, in a low number. Although the number of colonies does not allow any quantification, we noticed that in p140 cells their shape is smaller than in NeuT cells (Figure 1C for referee 1), suggesting that the stress condition of the anchorage-independent growth might differently affects p140 cells versus NeuT.

A

B

C

Figure 1, referee 1. Cell proliferation and anchorage-independent growth in primary NeuT and p140 cancer cells.

(A) Two populations for each genotype (NeuT-1, NeuT-2, p140-1, and p140-2 respectively), were seeded at 10^4 cells on 24 tissue culture dishes and left to proliferate for 4 days in the presence of medium supplemented with 10% FCS. Each day, cells were detached and manually counted in Burker chambers on triplicate wells.

(B) Protein extracts from cells at day 3 in culture were run on 6% SDS-PAGE and stained with the indicated antibodies.

(C) Anchorage independent growth by soft agar analysis of NeuT and p140 cells. 10×10^4 cells per 35-mm well, were plated in complete medium containing 0.35% agarose. After 4 weeks, colonies were methanol-fixed and stained with 0.005% crystal violet for visualization. Triplicate experiments were performed.

To further address this question, we analyzed the level of Cyclin D1 in protein extracts from day 12, 3D Matrigel:Collagen acini. As shown in the new Figure 4H, medium panel (here enclosed for reference convenience), p140Cap cells grown in 3D show a lower level of Cyclin D1 compared to NeuT cells, which is consistent with the observed decrease in acini area (Figure 4F) and with the decreased PCNA levels detected in p140-NeuT tumors in vivo (Figure 3E, see below for quantification).

H
Figure 4H. Primary p140Cap cancer cells restore mammary epithelial acina morphogenesis in 3D Matrigel-Collagen cultures

NeuT and p140 cells were plated on 3D Matrigel:Collagen I in a 1:1 ratio. On day 12, acinar structures were recovered and proteins extracted. Protein extracts were analysed by immunoblot and probed with antibodies to Cleaved Caspase 3, Cyclin D1 and Actin as loading control.

Overall, these findings indicate that the strong differences in tumor cell growth observed in Tg mice, in vivo xenotransplants and in 3D Matrigel:Collagen between NeuT and p140 cells may depend on a decreased growth rate of p140 cells, largely reliant on their different sensitivity to extracellular cues that come from an appropriate 3D environment.

Moreover, as also suggested by the reviewer, the observed defective growth of p140 cells in a 3D environment could be due to increased apoptotic rate. In tumors, we did not detect a significant variation in Caspase 3 expression (Figure 3E, see quantification, here enclosed for reference convenience).

E
Figure 3E. Quantification of PCNA and Activated Caspase 3 positive cells in tumor sections.

Paraffin-embedded sections from three NeuT and three p140-NeuT tumors taken from mice at 33 weeks of age were analyzed for Hematoxylin-Eosin H&E (a-d) and for immuno histochemistry with antibodies to NeuT (e-h), PCNA (i-n) and activated Caspase3 (o-r). Representative images are shown. Bar first and third columns: 50 microns. Bar second and fourth columns: 20 microns. Histograms on the left represent the percentage of PCNA or activated Caspase 3.

However, it is highly conceivable that the difference in total tumor burden reflects not only impaired tumor cell growth, but also the occurrence of local apoptotic events with remodeling of tumor structures, during tumor development. Therefore, we also analyzed the level of Cleaved Caspase 3 in protein extracts from day 12, 3D Matrigel:Collagen acini (see above, new Figure 4H, upper panel). In this context, p140Cap acini, consistent with the increased formation of an internal lumen highly reminiscent of normal mammary acina structures, are characterized by an increased level in Cleaved Caspase 3. Thus, at least in the 3D Matrigel:Collagen context, with a selective time point/kinetics of observations, we show that p140Cap can restore apoptotic pathways, typical of normal mammary epithelial cells during tissue morphogenesis (Mailleux et al., 2007).

We further analyzed apoptotic stimuli in 2D conditions. Indeed, p140Cap cells showed increased percentage of apoptotic cells, measured as positive Annexin 5 staining or increased expression of Cleaved Caspase 3, when subjected to apoptotic stimuli, such as starvation or matrix cell detachment (see Supplementary Figure 6B, here enclosed in part for referee's convenience).

B

Supplementary Figure 6B. p140Cap sensitizes the cells to apoptotic stimuli.

Left panels. The percentage of apoptotic cells were measured by FACS analysis (Annexin V and propidium iodide staining) in NeuT cells (red bars), and p140-NeuT cells (blu bars). Cells were subjected to 12 hours of starvation, or kept in suspension for 12 hours.

Middle panels. WB from protein extracts from NeuT/p140 cells treated as above, were probed with antibodies to Cleaved Caspase 3 and Tubulin as loading control.

Right panels. Quantification of Cleaved Caspase 3 expression in arbitrary Units (AU).

The data are the results of five independent experiments. Graph bars represent standard errors (* $p < 0.05$; ** $p < 0.01$).

Overall, our data indicate that p140Cap expression sensitizes the cells to apoptotic stimuli, thus supporting that p140Cap, in the context of ERBB2 tumor, has an inhibitory effect on 3D-dependent tumor growth both decreasing cell growth and promoting apoptosis. All these data have been discussed in the text.

There is lack of any experimentation and/or discussion of the impact of either CSK/SRC activation, or p130CAS in p140CAP effects towards ERBB2 signaling. SRC activation is critical for the transformed phenotype of activated ERBB2 and promotes Rac1 activation. Both p130CAS and TIAM1 are SRC substrates that eventually regulate Rac1 activity. This is also true for VAV2 another important Rac1 inducer in breast cancer cells. Given the prior findings of this group related to the role of p140CAP in SRC activation it is unclear why these relationships were not probed experimentally any further. ... As it stands, it is unclear whether this study relates to prior roles of p140CAP reported by the same group or not. While one may argue that novelty is somewhat reduced by further clarifying an already suggested mechanism of action, this reviewer believes this would add strength and not weakness to this story and help clarify the mechanism of p140CAP action.

We thank the reviewer for his/her comments. Indeed, we have already shown that p140Cap can control Src activation (Di Stefano et al., 2007, Damiano et al., 2010; Sharma et al., 2013). Interestingly, both in NeuT and in SKBR3 cells, p140Cap expression did not affect the activation of the Src kinase and the phosphorylation of its effector, paxillin and p130Cas (Supplementary Figure 9, here enclosed for referee's convenience). As positive control, we used MDA-MB-231, where over-expressed p140Cap down-regulates Src activation as well as p130Cas and paxillin phosphorylation. Taken together, these data indicate that in ERBB2 transformed cells, p140Cap does not impair Src activity, and acts through additional pathways for the control of cell migration.

All these data have been discussed in the text.

Supplementary Figure 9. p140Cap expression does not affect the activation of the Src kinase and the phosphorylation of its effectors, p130Cas and paxillin.

(A) Protein extracts from NeuT and p140 primary cancer cells were run on 6% SDS-PAGE and stained with antibodies to pSrc (Y416), Src, pp130Cas (Y410), p130Cas, pPaxillin (Y118) or Paxillin and GAPDH or Cofilin for loading control.

(B) Protein extracts from p140Cap over-expressing or mock SKBR3 cells were processed as in A.

(C) Protein extracts from p140Cap over-expressing or mock MD-MB-231 cells were processed as in A. p140Cap antibodies were used in the left panel. (* $p < 0.05$)

Reviewer #2 (Remarks to the Author): Expert on breast cancer mouse models

This manuscript deals with the observation that in 60% of HER2 amplified breast cancers, a p140Cap scaffold protein encoded by the SRCIN1 is also co-amplified. The authors provide evidence that amplification of SRCIN1 is correlated with good outcome. The authors show that overexpression of SRCIN1 in the mammary epithelium can reduce tumor burden in a MMTV-NeuT model. Mechanistically the authors argue that p140Cap suppresses tumor growth by interfering with TIAM activation of Rac 1. Although this is an interesting observation, the quality of data and the depth of analyses presented in this manuscript limit my enthusiasm for the paper.

We thank the reviewer for his/her comments and we apologize for the quality of the figures. This was indeed mostly due to the PDF conversion of our original files. We have now enclosed .jpeg images, but we would like to ask to the referee to be kind enough to go through the original Figures.

Specific Points

1. The FISH analyses shown in Fig, 2C is very poor quality (either the image or primary data) and thus is impossible to interpret.

We apologize with the referee: on the PDF version, images were not of sufficient resolution for properly assessing the data. We have now changed the panels in Figure 2C (here enclosed for referee's convenience), inserting higher magnification of the same samples (175 X).

C

SRCIN1 FISH

Fig. 2C. p140Cap FISH of breast cancer tissues.

Representative images of two cases of ERBB2 amplified tissues, labeled with a mix of two probes SRCIN1/CEP1; Red (SRCIN1) and green (CEP17) spots were automatically acquired at 175 X, using Metafer, by a MetaSystem scanning station. Panel a: 95% SRCIN1 amplification; average SRCIN1/nuclei= 11.7; panel b: 90% SRCIN1 amplification; average SRCIN1/nuclei= 13.4.

2. The characterization of the MMTV/ SRCIN1 strains is very cursory. There is no data shown to show the range of tissues the transgene is expressed. There is no immunohistochemical analyses demonstrating that SRCIN1 is expressed in mammary epithelial manner.

We apologize with the referee for not having enclosed data we already have obtained in the initial characterisation of the MMTV-p140Cap Tg mice. In the new Supplementary Figure 4, (here enclosed for referee's convenience), to the analysis of transgene expression in transgenic mice lines shown in panel A, we now add the expression of p140Cap in different tissues of Tg mice in panel B. Protein extracts from distinct tissues were collected from FVB WT or MMTV-p140Cap Tg female at 15 days of pregnancy. p140Cap transgene is expressed in mammary glands, parotid glands, lungs, and, at a minor extent, in brain.

Supplementary Figure 4A-B. Characterisation of MMTV-p140Cap Tg mice.

(A) Protein extracts from mammary gland at 3 days lactation from four independent Tg lines were analyzed. Samples were run on 6% SDS-PAGE and stained with antibodies to the Myc-Tag transgene or to vinculin, as loading control.

(B) p140Cap expression in different tissues of the MMTV-p140Cap transgenic mice. Protein extracts from distinct tissues collected from FVB WT or MMTV-p140Cap Tg female at 15 days of pregnancy, were run on 6% SDS-PAGE and stained with antibodies to p140Cap or to actin, as loading control. Representative images out of four different animals are shown.

We have also assessed the expression of p140Cap into mammary epithelial cells, as shown by IHC of Tg mammary glands at 3 days of lactation (Supplementary Figure 4D, included here for referee's convenience). All these data have been discussed in the text.

Supplementary Figure 4D. Characterisation of MMTV-p140Cap tg mice.

(D) p140Cap expression was evaluated by IHC on mammary glands from 3 day lactation transgenic mice.

3. Does ectopic expression of SRC1N1 impair normal mammary gland differentiation or development? This has direct bearing as the effect on mammary tumor induction indirect.

To address this point, mammary glands from WT and MMTV-p140Cap transgenic mice were collected at several stages of physiologic mammary development, namely at 6 weeks (puberal period) and 12 weeks (mature virgin mammary gland) of age, 12 days of pregnancy, 3 days of lactation and 5 days of post weaning involution. Five parameters were analysed to compare WT and Tg mammary glands, and reported in histograms (mean \pm SEM). No differences in TEB number, ductal elongation and ductal network area during mammary

gland growth and development in virgin mice, a non significant trend to lobular increase during pregnancy and lactation, and a small impairment in post weaning lobular involution in Tg mice were observed. Therefore, MMTV-p140Cap Tg mice do not show any defects in the development or the differentiation of the mammary gland that could impair tumor growth. All these data have been included in the new Supplementary Figure 4C (here enclosed for your convenience) and discussed in the text.

Supplementary Figure 4C. Characterisation of MMTV-p140Cap Tg mice during development and differentiation of the mammary glands.

(C) p140Cap expression in Tg mice does not impair mammary gland development and differentiation. Mammary glands from WT and MMTV-p140Cap transgenic mice were collected at several stages of physiologic mammary development, namely at 6 weeks (puberal period) and 17 weeks (mature virgin mammary gland) of age, 12 days of pregnancy, 3 days of lactation and 5 days of post weaning involution. Five parameters were analysed to compare WT and Tg mammary glands (see below), and reported in histograms (mean \pm SEM).. Bar: a, b, d, g, h, j 0,2 cm; c, e, i, k 300 μ m; f, l 20 μ m.

4. The quality of data showing apoptotic and proliferative capacity (Fig. 3E) is very poor. There is quantitative assessment of apoptotic and proliferative status-

We apologise with the reviewer. We have now quantified both PCNA and Caspase 3 staining, as shown in the new Figure 3E, histograms on the right (here enclosed for referee's convenience). PCNA quantification is $32,37 \pm 1,560$ in NeuT tumors versus $18,65 \pm 2,141$ in p140-NeuT tumors). Not significant differences were detectable in activated Caspase3 staining, in which only a few cells were positive in both tumor types ($7,694 \pm 2,257$ versus $7,381 \pm 2,408$) (panels o-r).

Figure 3E. Quantification of PCNA and activated Caspase 3 positive cells in tumor sections.

Paraffin-embedded sections from three NeuT and three p140-NeuT tumors taken from mice at 33 weeks of age were analyzed for Hematoxylin-Eosin H&E (a-d) and for immuno histochemistry with antibodies to NeuT (e-h), PCNA (i-n) and activated Caspase3 (o-r). Representative images are shown. Bar first and third columns: 50 microns. Bar second and fourth columns: 20 microns. Histograms on the right show the percentage of PCNA+ (upper panel) and Activated Caspase3+ (lower panel) cells. Statistical significant differences were evaluated using unpaired t tests (**p < 0.001).

5. The authors should also examine whether tumor impairment reflects the level of angiogenic infiltration (CD31 staining)

6. The authors should stain the tumor sections with CK8, CK14 and CK5 to establish whether the luminal status of the tumors is altered.

To address these points, we have assessed the level of angiogenic infiltration by CD31 marker staining in tumor sections. Angiogenic infiltration was decreased in p140 tumors ($9648 \pm 351,5$ versus $5344 \pm 232,8$) (see Supplementary Figure 5B, here enclosed for referee's convenience). These data have been discussed in the text.

Moreover, as suggested by the reviewer, we have stained tumor sections from p140-NeuT mice with antibodies to keratins. As shown in the Supplementary Figure 5A (here enclosed for referee's convenience), in p140-NeuT tumors a normal residual duct is outlined by a continuous layer of CK5 positive basal cells while the whole surrounding tumor is completely negative. On the contrary, both luminal duct cells and the whole tumor are positive for the CK8-18 staining, indicating that p140Cap expression does not alter the luminal status of the tumor.

A

B

Supplementary Figure 5. Characterization of luminal status and angiogenic infiltration in NeuT and p140 tumors

(A) Formalin fixed paraffin embedded sections from three NeuT and three p140-NeuT tumors were analyzed for immunohistochemistry with antibodies to CK5 and CK8-18. Bar on the left panels 200 microns. Bar on the right panels 30 microns.

(B) Cryo-section from three NeuT and three p140-NeuT tumors taken from mice at 33 weeks of age were analyzed for immunohistochemistry with antibodies against endothelial cells: CD31 and CD105. Representative images are shown. Bar 200 microns. Histogram indicates the number of red pixels counted in 6 low power fields per tumor.

5. Again the quality of data shown in Figure 4H is poor.

This was indeed mostly due to the PDF conversion of our original files. We have now enclosed .jpeg images, but we would like to ask to the referee to be kind enough to go through the original Figures.

7. The effects of ectopic expression of SRC1N1 on the spontaneous metastasis should be included in the analyses.

We thank the reviewer for this comment, which was not easy to address due to the model, but at the end gave substantial data. Indeed, we have already analysed metastasis formation in Tg mice. At the time when mice need to be suppressed according to the rules of our animal house (when at least one tumor was above 1 cm diameter), no metastasis were found. Tumor resection was not possible because most of the 10 glands were bearing a tumor. We moved to spontaneous metastasis from orthotopic tumors in NSG mice, where cells were injected, tumors resected at 1 cm diameter, and organs analysed 5 weeks later. No metastasis were found when primary NeuT cells were used, indicating that this model was not aggressive enough to allow cells to move out spontaneously from the primary tumors. To address this point, we moved to the NeuT-TUBO cells, an additional transplantable primary NeuT cell model from the BALB/c background (Rovero et al., 2000). Upon infection with empty or p140Cap retroviruses, we generated NeuT-TUBO (as mock cells), and p140-TUBO cells (Supplementary Figure 7A, here enclosed for referee's convenience). We showed that p140Cap expression significantly limited tumor cell growth upon transplantation (Supplementary Figure 7B).

Supplementary Figure 7. Characterisation of NeuT-TUBO and p140-TUBO cells

(A) NeuT-TUBO p140-TUBO cells obtained by stable infection with empty pBABE or p140Cap-pBABE retroviruses. The p140Cap over-expression was checked by western blot analysis using antibodies to NeuT, p140Cap and GAPDH, as loading control.

(B) 5×10^5 cells as in (A) were injected in the right fat pads of nude mice. Tumor diameters were measured every two days for 26 days. Two independent experiments were performed using 5 mice per group. Differences in tumor volume were evaluated using two-way ANOVA followed by Bonferroni multiple comparison post hoc tests (**p < 0.001).

In the experimental metastasis assays upon tail vein injection, NeuT-TUBO cells gave rise, after 25 days, to numerous large lung metastases substituting approximately 80% of lung

tissue area. At the same time point, p140-TUBO cells were grown to occupy only about 54% of lung tissue area (Figure 6A, here enclosed for referee's convenience). Since this assay is only a proxy to measure the metastatic potential of cell, we moved to the spontaneous metastasis assay from primary tumors. As shown in Figure 6B, , here enclosed for referee's convenience, tumors originated from p140-TUBO cells gave rise to a significant reduced number of lung metastases over tumors grown from NeuT-TUBO cells. Moreover, the immunohistochemical staining with anti NeuT/ERBB2 and anti p140Cap antibodies of lungs with p140-TUBO metastases showed that while all the metastases were positive for the NeuT protein, p140Cap strong expression was conserved only in small metastases (Figure 6C). Interestingly, larger metastases expressed only low levels of p140Cap and, accordingly, showed a less nodular histological structure similar to those developed from NeuT-TUBO tumors (Figure 6D, compare upper and lower panels). Overall, these data indicate that p140Cap counteracts metastasis formation.

Figure 6. p140Cap impairs spontaneous metastasis

(A, B) Representative gross observation (panel a, b), Hematoxylin&Eosin sections (panel c, d) and quantitative analysis in the experimental lung metastasis assay (A) and in the spontaneous lung metastasis assay (B).

In A, 5×10^4 cells were injected into the tail vein of NSG mice. After 4 weeks, the lungs were explanted and analyzed. Two independent experiments were performed using 5 mice per group. The histogram shows the percentage of metastatic lung tissue on total lung area. Statistical significance was evaluated with unpaired t-test (** $p < 0.001$). Error bar: SEM.

In B, 10^5 cells were injected in the right fat pads of NSG mice. Tumor volume were measured every week till the tumors reached a 10 mm diameter, when tumors were surgically removed. After 5 weeks, mice were sacrificed and lungs were explanted. The histogram shows the number of lung metastasis. Statistical significance was evaluated with unpaired t-test ($*p < 0.1$). Error bar: SEM. Scale bar A-B: 800 micron.

(C) NeuT (panel a, c) and p140Cap expression (panel b, d) in spontaneous lung metastasis of mice injected with NeuT-TUBO cells (panel a, b) and mice injected with p140-TUBO cells (panel c, d). Alternate consecutive sections were stained. Scale bar: 400 micron

(D) High magnification fields of rectangular areas in Figure C, panel d. Scale bar: 50 micron.

Moreover, to further characterising the NeuT-TUBO cells, we performed Rac and Tiam1 activity assays, as well as co-immunoprecipitation assay. p140-TUBO cells show decreased Rac and Tiam1 activity, and p140Cap co-immunoprecipitates with Tiam1 in these cells (see new Supplementary Figure 11, here enclosed for your convenience).

Supplementary Figure 11. Tiam1/Rac axis in NeuT-TUBO and p140-TUBO cells

(A) Active Rac pull down from NeuT-TUBO and p140-TUBO cells. Eluted material (upper panels) and cell extracts (lower panels) run on 12% SDS-PAGE revealed with anti Rac antibodies. Histograms show the ratio between active and total Rac protein levels in arbitrary units (A.U.) from four independent experiments. Statistical significant differences were evaluated using unpaired t tests ($*p < 0.05$; $**p < 0.01$). Error bar: SEM.

(B) Extracts from NeuT-TUBO and p140-TUBO cells were immunoprecipitated with antibodies to p140Cap (upper panels) or Tiam1 (lower panels). Cell extracts and immunoprecipates were run on 6% SDS-PAGE and blotted with antibodies to p140Cap and Tiam1. Representative images from five independent experiments are shown.

(C) The level of active Tiam1 was determined using the active Rac-GEF assay kit in NeuT-TUBO and p140-TUBO cells. Equal amount of extracts were incubated for 1h at 4°C with Rac G15A agarose beads. Active Tiam1 and total Tiam1 levels were determined using an anti-Tiam1 antibody for western-blot detection, from eluted material and input fractions, respectively. The histogram represents the quantification of active Tiam1 in three independent experiments, normalizing active Tiam1 levels to the corresponding total Tiam1 levels in arbitrary units (A.U.).

8. Figure 5 should include immunohistochemical analyses of E-cadherin staining in addition to immunoblot

We thank the reviewer for this comment. In the new Figure 5D (here enclosed for your convenience), we have included two representative pictures of E-cadherin staining by immunofluorescence on tumor sections that confirmed the increased expression of E-cadherin at the cell membrane in p140 tumors compared to NeuT tumors.

Figure 5D: E-cadherin staining on tumor sections

Paraffin-embedded sections from three NeuT and three p140-NeuT tumors taken from mice at 33 weeks of age were analyzed for immunofluorescence with antibodies to E-Cadherin (green). Representative images are shown. The scale bar represents 50 microns. Image acquisition was performed using Zeiss LSM 510 META confocal microscope. The E-cadherin mean fluorescence intensity was evaluated on the digital images of three tumors per group (4 X 200 microscopic fields per sample) with ImageJ, using the Mean Gray Value.

9. *Figure 5E is of very poor quality and should be removed or repeated.* According to the referee's comment, we have removed panel E.

10. *Figure 6G and H are of very poor quality (Indeed only small selected fields are shown)*

We really apologize with the referee. This was indeed mostly due to the PDF conversion of our original files. We have now enclosed .jpeg images, but we would like to ask to the referee to be kind enough to go through the original Figures.

11. *The authors should inactivate SRC1N1 in a number of ErbB2 amplified breast cancer lines rather than the single SKBr3 line shown.*

To address this point, we choose the ERBB2 positive MDA-MB-453 cells, obtained from ATCC. As assessed by qRT-PCR, these cells are equivalent to SKBR3, since they display amplification of the ERBB2 gene, but not of the SRCIN1 gene. These cells express p140Cap protein and we were able to silence its expression at the same extent as the SKBR3 cells (new Supplementary Figure 8C). As shown in the new Figure 7, panels E, F, p140Cap silencing in MDA-MB-435 led to increased cell migration and Rac activation. Consistently, as shown in the new Figure 8F, Tiam1 activity is increased in p140Cap silenced MDA-MB-453 cells, as in the SKBR3 cells.

We have collected all the data here for referee's convenience in the Figure 1 for referee 2.

Figure 1 for referee 2: p140Cap silencing in MDA-MB-453 cells increases cell migration, and Rac/Tiam1 activation.

(A) MDA-MB-453 cells were transiently transfected with ON-TARGET plus human SRCIN1 small-interfering RNA (siRNA) (si p140) or ON-TARGET plus non-targeting siRNA (Dharmacon RNAi) (si ctrl). p140Cap over-expression was checked by western blot analysis using antibodies to p140Cap and GAPDH, as loading control. Quantification of four distinct experiments is shown in arbitrary units on the left.

(B) Representative images of Transwell migration assays. 10^5 cells were left to migrate for 24 h in the presence or the absence of 15% FBS, fixed, stained and counted. Histograms represent on the y-axis the fold increase (ratio between the number of cells migrated in the presence and in the absence of FBS), from three independent experiments, performed in triplicate. Error bar: SEM.

(C) Active Rac pull down. Eluted material (upper panels) and cell extracts (lower panels) run on 12% SDS-PAGE revealed with anti-Rac antibodies. Histograms show the ratio between active and total Rac protein levels in arbitrary units (A.U.) from five independent experiments. Statistical significant differences were evaluated using unpaired t tests (* $p < 0.05$; ** $p < 0.01$). Error bar: SEM.

(D) The level of active Tiam1 was determined using the active Rac-GEF assay kit in MDA-MB-453 (si p140) cells. Equal amount of extracts were incubated for 1 h at 4°C with Rac G15A agarose beads. Active Tiam1 and total Tiam1 levels were determined using an anti-Tiam1 antibody for western-blot detection, from eluted material and input fractions, respectively. Antibodies to tubulin and GAPDH were used as loading controls. The histogram represents the quantification of active Tiam1 in three independent experiments, normalizing active Tiam1 levels to the corresponding total Tiam1 levels in arbitrary units (A.U.).

Minor Comments

The reference to Neu T strain used is not a primary reference. Because several strains of this type were generated that may have different properties this is important point.

We thank the reviewer for this observation. We now cite the primary references in the text as refs 25-28.

Throughout the manuscript, blots comprise 2 or 3 samples at most. An inclusion of at least 5 samples for each experimental point would increase the confidence in the validity of the data.

The western blots that have been included in this manuscript are consistent and reproducible data, from many different experiments, extensively repeated more than five times in my lab. For a sake of simplicity, the quantification have always been performed on at least three distinct experiments. We went back to the experiments and we included additional data in some Figures.

Many of immunofluorescence images shown are not of publication quality.

We really apologize for this inconvenience and we hope that we reached a higher quality in this revised version.

REVIEWERS' COMMENTS:

Reviewer #1 (Remarks to the Author):

First of all, the revised manuscript by Grasso et al. is significantly improved. The authors have generated new data to address my main concerns, which centered primarily around the possibility that all observed effects of p140cap could be explained by potential effects on tumor cell growth and apoptosis. The data provided make it clear that expression of p140cap affects both the growth rate and the apoptosis of Her2 overexpressing breast cancer cells. My main criticism of the current version is that it still is focused at least in the introduction and discussion on the effects of p140cap on migration/invasion and metastasis. Frankly, I am not convinced by any of the data provided that such effects exist:

The model of tail vein metastasis is known to be affected by extravasation and homing, but also by conditions that regulate anchorage-independent growth, or apoptosis.

The orthotopic metastasis model cannot be evaluated properly because the tumors do not grow at the same rate. Instead, the presence of metastases expressing high levels of p140cap suggests that it is not inhibiting metastatic spread. However, the loss of p140cap expression in the larger tumors indicates again that the main effect of p140cap is on tumor growth.

Similar concerns exist for the migration experiments, which were performed over 24 hours, and would be significantly affected by the presence of any immobile apoptotic cells.

The 3D in vitro and tumor morphological effects can be easily explained by the observed changes in tumor and EMT markers, as well as the effects of Rac1 on apoptosis.

In summary, it is clear from the data provided that p140cap plays a role in breast cancer progression. The animal data in general support the human IHC/survival data, although the MMTV-Neu mouse is not considered an accurate model of human Her2+ breast cancer. The observed effects are probably due to overall changes in tumor cell growth and the data support involvement of Rac1 signaling. This is not a novel mechanism of Her2 action though, nor it suggests novel means of targeting Her2+ breast cancer, and observed effects of p140cap overexpression in the animal models are rather minimal.

Reviewer #2 (Remarks to the Author):

The revised version is improved and address most of the raised concerns.

Rebuttal letter to REVIEWERS' COMMENTS:

Reviewer #1 (Remarks to the Author):

First of all, the revised manuscript by Grasso et al. is significantly improved. The authors have generated new data to address my main concerns, which centered primarily around the possibility that all observed effects of p140cap could be explained by potential effects on tumor cell growth and apoptosis. The data provided make it clear that expression of p140cap affects both the growth rate and the apoptosis of Her2 overexpressing breast cancer cells.

We thank the reviewer for his/her positive comments on the revised version.

My main criticism of the current version is that it still is focused at least in the introduction and discussion on the effects of p140cap on migration/invasion and metastasis. Frankly, I am not convinced by any of the data provided that such effects exist.

We are sorry for this inconvenience. We have now tried to address these concerns modifying the Abstract, Results and Discussion to better presenting the concept that p140Cap dampens tumor progression both down-regulating tumor cell growth and enhancing apoptosis, and interfering with dissemination and local growth of metastases in specific organs.

In the Abstract (see page 3) we have modified a sentence, introducing the ability of p140Cap to counteract tumor growth.

p140Cap dampens ERBB2 tumor cell progression, impairing tumor onset and growth in the NeuT mouse model, and counteracting epithelial mesenchymal transition, resulting in decreased metastasis formation. One major mechanism is the ability of p140Cap to interfere with ERBB2-dependent activation of Rac GTPase-controlled circuitries.

In the Introduction, we have underlined the recent literature showing a general role of p140Cap on proliferation and migration of tumor cells (see page 4). These papers were already present in the Discussion of the previous revised version, but now we moved this concept earlier, to strengthen the role of p140Cap in tumor cell proliferation and migration.

We have previously described the p140Cap adaptor protein as a molecule that interferes with adhesion properties and growth factor-dependent signaling, thus affecting tumor features in cancer cells^{16, 17, 18, 19}. ***Recent reports have underlined that p140Cap regulates proliferation and migration in colon, lung, gastric, cutaneous squamous carcinoma and osteosarcoma cells***^{19, 20, 21, 22, 23, 24}

Moreover, we have changed the last sentence of the Introduction (see page 5).

Altogether, our results argue for a key role of p140Cap in curbing the aggressiveness of the ERBB2 tumors, ***counteracting in vivo tumor growth***, epithelial mesenchymal transition ***and metastatic lesions***.

In the Results, we have modified a sentence at page 23, to underline the ability of p140Cap to affect tumor growth and sensitivity to apoptosis.

The sum of results from i) the analysis of the clinical cohort (reduction of metastatic risk in ERBB2 tumors), ii) the experiments performed in vivo (***reduced tumor masses***, reduced "metastatic" ability and decreased expression of EMT markers in the animal model), and iii) in vitro (reduced

protrusive ability of acini and restoration of polarity in the 3D-morphogenetic assays), points to a counteraction of p140Cap on ERBB2-dependent tumor progression. ***All these data show that p140Cap dampens tumor features, affecting tumor growth, sensitivity to apoptosis and metastatic properties of cancer cells.*** In search

We worked extensively on the Discussion, also introducing additional references (58, 59).

At page 26:

The clinical evidence that p140Cap correlates with a favorable outcome in ERBB2 breast cancer patients suggest that p140Cap is able to curb the intrinsic biological aggressiveness of ERBB2 tumor (Figure 8G). ***Indeed, p140Cap confers to ERBB2 transformed cells limited in vivo tumor growth ability and spontaneous lung metastasis formation.*** This less aggressive phenotype is likely linked to ***reduced cell proliferation, assessed by a decreased staining of the proliferative marker PCNA in tumors, increased sensitivity to apoptosis, and*** strong inhibition in the EMT program observed in p140Cap expressing tumor cells.

At pages 27/28:

Notably, p140Cap expression significantly limits the ability of ERBB2 transformed cells to give rise to metastasis, both in experimental and in spontaneous metastasis assays. Indeed, ***when comparing tumors of the same size, p140Cap tumors give rise to a significantly lower number of spontaneous lung metastasis compared to NeuT tumors, suggesting that p140Cap affects metastatic spread. However, when analyzing the metastatic lesions from p140Cap tumors, we observed a strong p140Cap expression only in smaller lung metastasis, rather than in larger ones, suggesting that p140Cap has also a strong effect on local metastatic growth. Therefore, from these data we can conclude that p140Cap significantly impairs metastasis acting both on tumor cell spreading and on metastatic growth, due to its ability to down-regulate tumor cell growth and to enhance apoptotic events.***

Rac GTPase is a well-known mediator of human ERBB2 breast cancer progression^{45, 46, 47, 48}, ***affecting signaling pathways impinging on tumor cell proliferation, apoptosis and acinar structure⁵⁸, as well as metastasis dissemination⁵⁹. Here we show that p140Cap strongly impairs Rac activation in both human and mouse ERBB2 transformed cells. Indeed, in 3D Matrigel-Collagen morphogenetic assay, the Rac inhibitor NSC23766⁵⁰ consistently decreased the area of the NeuT organotypic structures and restored cell polarity disrupted by the oncogene, thus recapitulating the effect of p140Cap expression. Of note, expression of a constitutively active form of Rac in p140Cap cells was able to rescue the aggressive ERBB2 phenotype, increasing acinar area and decreasing the percentage of polarized structures.*** These results further point to the mechanistic relevance of p140Cap/Rac counteraction as an essential step for limiting ERBB2 tumor progression.

The model of tail vein metastasis is known to be affected by extravasation and homing, but also by conditions that regulate anchorage-independent growth, or apoptosis.

The orthotopic metastasis model cannot be evaluated properly because the tumors do not grow at the same rate. Instead, the presence of metastases expressing high levels of p140cap suggests that it is not inhibiting metastatic spread. However, the loss of p140cap expression in the larger tumors indicates again that the main effect of p140cap is on tumor growth.

We thank the reviewer for this comment. In fact, we realized that, in the revised version, we did not point out that the orthotopic spontaneous metastasis assays were all performed by comparing NeuT and p140 tumors of the same size. This allowed us to reasonably exclude the trivial effect of growth on the number of metastases.

We include this essential information in the Results (see page 22),

Since this assay is only a proxy to measure the metastatic potential of cells, we moved to the spontaneous metastasis assay from primary tumors, ***comparing metastasis formation from tumors of the same size.***

and in the Discussion as well (see above at page 26).

The 3D in vitro and tumor morphological effects can be easily explained by the observed changes in tumor and EMT markers, as well as the effects of Rac1 on apoptosis. In summary, it is clear from the data provided that p140cap plays a role in breast cancer progression. The animal data in general support the human IHC/survival data, although the MMTV-Neu mouse is not considered an accurate model of human Her2+ breast cancer. The observed effects are probably due to overall changes in tumor cell growth and the data support involvement of Rac1 signaling. This is not a novel mechanism of Her2 action though, nor it suggests novel means of targeting Her2+ breast cancer, and observed effects of p140cap overexpression in the animal models are rather minimal.

We have introduced additional papers on Rac GTPase involvement in ERBB2 tumors, which support its role in sustaining signaling involved in cell proliferation, apoptosis and migration (ref 58, 59). However, the fact that the Rac GTPase has an essential role in the ERBB2 breast cancers, in our opinion, strengthen our data, which introduce p140Cap as a new regulator of Rac activity at the level of its GEF Tiam1.

Reviewer #2 (Remarks to the Author):

The revised version is improved and address most of the raised concerns.

We thank the reviewer for the positive comments on the revised version.